# Evaluation of modeled summertime convective storms using polarimetric radar observations

Prabhakar Shrestha[1], Silke Trömel[1,2], Raquel Evaristo[1], and Clemens Simmer[1]

[1]Institute of Geosciences, Meteorology Department, Bonn University, Germany
[2]Laboratory for Clouds and Precipitation Exploration, Geoverbund ABC/J, Bonn, Germany

**Correspondence:** Prabhakar Shrestha (pshrestha@uni-bonn.de)

**Abstract.** Ensemble simulations with the Terrestrial Systems Modelling Platform (TSMP) covering north-western Germany are evaluated for three summertime convective storms using polarimetric X-band radar measurements. Using a forward operator, the simulated microphysical processes have been evaluated in radar observation space. Observed differential reflectivity ($Z_{\mathrm{DR}}$) columns, which are proxies for updrafts, and multi-variate fingerprints for size sorting and aggregation processes are captured by the model, but colocated specific differential phase ($K_{\mathrm{DP}}$) columns in observations are not reproduced in the simulations. Also, the simulated $Z_{\mathrm{DR}}$ columns, generated by only small-sized supercooled drops, show smaller absolute $Z_{\mathrm{DR}}$-values and a reduced width compared to their observational counterparts, which points to deficiencies in the cloud microphysics scheme as well as the forward operator, which does not have explicit information of water content of ice hydrometeors. Above the melting layer, the simulated polarimetric variables also show weak variability, which can be at least partly explained by the reduced particle diversity in the model and the inability of the T-matrix method to reproduce the polarimetric signatures of snow and graupel; i.e. current forward operators need to be further developed to fully exploit radar data for model evaluation and improvement. Below the melting level, the model captures the observed increase in reflectivity, $Z_{\mathrm{DR}}$ and specific differential phase ($K_{\mathrm{DP}}$) towards the ground.

The contoured frequency altitude diagrams (CFADs) of the synthetic and observed polarimetric variables were also used to evaluate the model microphysical processes statistically. In general, CFADs of the cross-correlation coefficient ($\rho_{\mathrm{hv}}$) were poorly simulated. CFADs of $Z_{\mathrm{DR}}$ and $K_{\mathrm{DP}}$ were similar but the model exhibits a relatively narrow distribution above the melting layer for both, and a bi-model distribution for $Z_{\mathrm{DR}}$ below the melting layer, indicating either differences in the mechanism of precipitation formation or errors in forward operator which uses a functional form of drop size distribution.

In general, the model was found to underestimate the convective area fraction, high reflectivities, and the width/magnitude of $Z_{\mathrm{DR}}$ columns, all leading to an underestimation of the frequency distribution for high precipitation values.

## 1 Introduction

The representation of cloud and precipitation processes in atmospheric models is a central challenge for numerical weather prediction and climate projections (e.g. Boucher et al., 2013; Bauer et al., 2015). Especially, the parameterization of cloud microphysical processes and its interaction with the resolved dynamics need to be well tuned in order to provide dependable

predictions (Igel et al., 2015; Brown et al., 2016; Morrison et al., 2020). In numerical models, the cloud microphysics is parameterized either using the so-called spectral (bin) approach or single/multi-moment bulk formulations, with the latter most common in numerical weather prediction (NWP) models due to computational efficiency (Khain et al., 2000). These parameterizations are often constrained using in-situ and/or radar reflectivity observations. While in-situ measurements by aircrafts are sparse, ground-based radar observations provide three-dimensional structure of microphysical processes and are thus increasingly used for in-depth numerical modelling evaluation (e.g. Noppel et al., 2010; Min et al., 2015; Tao et al., 2016, of many others). Besides horizontal reflectivity $Z_H$, polarimetric radar observations provide estimates of differential reflectivity $Z_{DR}$, specific differential phase $K_{DP}$, and cross-correlation coefficient $\rho_{hv}$, which depend on hydrometeor shape, orientation, density and phase composition, and thus enable a more detailed evaluation of the modeled microphysical and macrophysical processes (Andrić et al., 2013; Snyder et al., 2017a; Putnam et al., 2017). However, this research field is still relatively new, partly because polarimetric precipitation radar networks became just recently available. The upgrade of the United States National Weather Service (NWS) S-band Weather Surveillance Radar 1988 Doppler (WSR-88D) network to polarimetry was completed in 2013, while Germany completed the upgrade of its national C-band network in 2015 in parallel with other European countries.

Measured polarimetric variables are the result of the average scattering characteristics of the ensemble of hydrometeors contained in a resolved radar resolution volume, and are expressed as second order moments or correlations and powers of the horizontally and vertically polarized signals (Ryzhkov and Zrnic, 2019). Polarimetric variables are affected by hydrometeor shape/size distribution, concentration, orientation and phase composition, but all to a different extent and therefore the multi-variate fingerprints provides insights into various microphysical processes like size sorting, evaporation, aggregation, riming, melting, secondary ice production, hail production etc. Horizontal reflectivity ($Z_H$) especially provides information on the size and with that on ongoing aggregation/riming processes. Differential reflectivity ($Z_{DR}$) mainly provides information on the shape of hydrometeors and does not depend on the number concentration, while specific differential phase ($K_{DP}$) is proportional to the concentration of hydrometeors, thereby providing insight into the generation of new snow in the dendritic growth layer (Trömel et al., 2019). Cross-correlation coefficient ($\rho_{hv}$) is mainly a measure of the hydrometeor diversity in the resolved radar resolution bin. This information can be used for numerical model evaluation using two approaches: (1) the comparison of simulated mixing ratios or process rates with microphysical and thermodynamic retrievals from radar observations and (2) the direct comparison in radar observation space exploiting synthetic measurements obtained from a forward operator (Ryzhkov et al., 2020; Trömel et al., 2021). While both approaches have uncertainties caused by inherent assumptions, the latter method recently received more attention in the community due to increasingly available forward operators (e.g. Pfeifer et al., 2008; Xie et al., 2016; Heinze et al., 2017; Wolfensberger and Berne, 2018; Kumjian et al., 2019; Matsui et al., 2019; Oue et al., 2020), but requires awareness of assumptions made in both the model and the forward operator (FO). Even though first polarimetric forward operators have been already available several years ago, like SynPolRad introduced in Pfeifer et al. (2008), refinements are still ongoing and mandatory for a full exploitation. E.g. Shrestha et al. (2022) and Trömel et al. (2021) demonstrated the limitations of the T-matrix method and its assumption of oblate spheroids used in current forward operators to reproduce the polarimetric signatures of low density particles like dry snow aggregates, and motivated further research towards a full

exploitation of radar observations for model evaluation. The connection to a scattering data base would be key for a better representation of the ice phase. Furthermore, several other key tools became just recently available or are still under development (Trömel et al., 2021). Besides, many previous studies have documented polarimetric signatures of deep convective storms in S-band or C-band observations (e.g. Kumjian and Ryzhkov, 2008; Jung et al., 2010, 2012; Kumjian and Ryzhkov, 2012; Homeyer and Kumjian, 2015; Kaltenboeck and Ryzhkov, 2013; Johnson et al., 2016; Ilotoviz et al., 2018), while studies based on higher resolved X-band measurements with more pronounced signals in $K_{\mathrm{DP}}$ are still gaining grounds (Kim et al., 2012; Snyder et al., 2010, 2013, 2017a; Figueras i Ventura et al., 2013; Suzuki et al., 2017; Allabakash et al., 2019; Das et al., 2021; Trömel et al., 2021).

As an ongoing effort on the fusion of models and radar polarimetry, this study focuses on the evaluation of a soil-vegetation-atmosphere modeling system, using polarimetric observations from X-band radar. The Terrestrial Systems Modelling Platform (TSMP; Shrestha et al., 2014; Gasper et al., 2014) was developed to better represent biogeophysical processes in regional coupled atmosphere-landsurface models with explicit representation of surface groundwater interactions and to eventually improve modeled land-atmosphere interactions and system state predictions (Simmer et al., 2015). TSMP has been extensively evaluated over north-western Germany for hydrological processes and land-atmosphere interactions (Shrestha et al., 2014; Rahman et al., 2015; Sulis et al., 2015; Uebel et al., 2017; Shrestha, 2021a). So far, however, polarimetric radar observations, which offer in-depth information on clouds and precipitation microphysical composition and evolution, have not yet been exploited for the evaluation of the modelling platform. Therefore, the main goal of this study is to extend TSMP with a forward operator and to perform km-scale ensemble simulations in convection permitting mode, to evaluate 2-moment cloud microphysics scheme (Seifert and Beheng, 2006) for multiple convective storms with attenuation corrected high resolution X-band polarimetric radar data. The 2-moment scheme allows the possibility of aerosol-cloud-precipitation interaction studies and hence the possibility of understanding aerosol effects on polarimetric quantities. Importantly, the 2-moment cloud microphysics scheme is also a candidate for the Icosahedral Nonhydrostatic Weather and Climate Model (ICON; Zängl et al., 2015) used for operational weather forecasting by Deutscher Wetterdienst (DWD, Germany). We make an effort to explore the prominent polarimetric features of the observed convective storms, examine whether these features are adequately captured by the model, and also evaluate whether the model is able to capture the observed statistical properties of the polarimetric variables.

The manuscript is structured as follows: Section 2 describes the model and polarimetric radar forward operator. The polarimetric radar observations are presented in Sect. 3. The experiment setup is described in Sect. 4. Results of model evaluation in radar space, including the comparison with radar based precipitation estimates are presented in Sect. 5. Discussion and conclusions are provided in Sect. 6 and 7 respectively.

## 2 Model and Forward Operator

### 2.1 Model

The Terrestrial Systems Modelling Platform (TerrSysMP or TSMP; Shrestha et al. 2014; Gasper et al. 2014; Shrestha and Simmer 2020) connects three models for the soil-vegetation-atmosphere continuum using the external coupler OASIS3-MCT

(Craig et al., 2017). The soil-vegetation component consists of the NCAR community Land Model CLM3.5 (Oleson et al., 2008) and the 3D variably saturated groundwater and surface water flow model ParFlow (Jones and Woodward, 2001; Ashby and Falgout, 1996; Kollet and Maxwell, 2006; Maxwell, 2013). The atmospheric component consists of the operational German weather forecast model COSMO (Consortium of Small-scale Modelling; Doms and Schättler 2002; Steppeler et al. 2003; Baldauf et al. 2011). The dynamical core of COSMO uses the two time-level, third order Runge–Kutta method to solve the compressible Euler equations (Wicker and Skamarock 2002; Baldauf et al. 2011). The equations are formulated in a terrain-following coordinate system with variable discretization using the Arakawa C-grid. The physical packages used in this study are the radiation scheme based on the one-dimensional two-stream-approximation of the radiative transfer equation (Ritter and Geleyn, 1992), a shallow convection scheme based on (Tiedtke, 1989), a 2-moment bulk microphysics scheme (Seifert and Beheng 2006, hereafter referred as SB2M) and a modified turbulence level 2.5 scheme of Mellor and Yamada (1982)(Raschendorfer, 2001). We discuss the cloud microphysics scheme relevant for this study in more detail below; more detailed discussions of the dynamical and physical processes in COSMO can be found in Baldauf et al. (2011).

SB2M is used in an extended version with a separate hail class (Blahak, 2008) and a new cloud droplet nucleation scheme based on lookup tables (Segal and Khain, 2006) and raindrop size distributions with the shape parameter dependent on the mean diameter for sedimentation and evaporation (Seifert, 2008; Noppel et al., 2010). SB2M predicts the mixing ratios ($q_x$) and number densities ($N_x$) of cloud droplets, rain, cloud ice, snow, graupel and hail particles, which are all assumed to follow a generalized Gamma distribution,

$$f(x) = Ax^\nu \exp(-\lambda x^\mu) \tag{1}$$

where x is the mass of the hydrometeor and $A$, $\mu, \nu$ and $\lambda$ are the intercept, spectral shape and slope parameters, respectively. While the shape parameters are prescribed, $A$ and $\lambda$ can be estimated using the zeroth and the first moments of the distribution. The equivalent/maximum diameter ($D_x$) of spherical/non-spherical hydrometeors is given by

$$D_x = ax^b \tag{2}$$

The shape parameters of the Gamma distribution (Eq. 1) and and power law relationship between diameter and particle mass (Eq. 2) for different hydrometeors used in this study are summarized in Table 1. Further, SB2M does not have a prognostic melted fraction, and instantaneously transfers the amount of meltwater formed during one model timestep from cloud ice, snow, graupel, and hail to the rain class.

The activation of CCN from aerosols in SB2M is based on pre-computed activation ratios stored in a lookup table (Seifert et al., 2012), which depend on the vertical velocity and background aerosol properties (Segal and Khain, 2006). The aerosol is assumed to be partially soluble with a two mode lognormal size distribution. This requires the specification of the condensation nuclei (CN) concentration, the mean radius of the larger aerosol mode, the logarithm of its geometric standard deviation, and its solubility. The vertical profile of the CN concentration is assumed constant up to 2 km height followed by an exponential decay above. The ice nuclei (IN) number densities of dust, soot and organics are also prescribed for heterogeneous ice nucleation based on the parameterization of Kärcher and Lohmann (2002) and Kärcher et al. (2006). Table 2 summarizes the large-scale

aerosol specification for the cloud droplet and ice particle nucleation used in this study. In absence of an prognostic aerosol model, the prescribed values remain constant, and processes like scavenging or chemical transport are not modeled.

## 2.2 Forward Operator

The Bonn Polarimetric Radar forward Operator (B-PRO; Heinze et al., 2017; Xie et al., 2021; Trömel et al., 2021; Shrestha et al., 2022) used in this study is a polarimetric extension of the non-polarimetric EMVORADO (Zeng et al., 2016) operator, which computes the polarimetric radar variables from scattering amplitude calculations using the T-matrix method (Mishchenko et al., 2000). The synthetic polarimetric moments are ouput on the spatial grid given by the numerical model field.

B-PRO simulates the polarimetric radar variables at specified weather radar wavelengths (X-band—3.2 cm) using prognostic model states of temperature, pressure, humidity, wind velocity, mixing ratio and number densities of hydrometeors. Besides cloud liquid class, the hydrometeors are interpreted as homogeneous oblate spheroids in the T-matrix computation. Additional uncertainties in the polarimetric estimates arise from required hydrometeor information usually not available from the model like spheroid diameter ($D_x$), aspect ratio ($AR$), width of canting angle distributions $\sigma_c$, and dielectric constant. The latter is further dependent on hydrometeor density, water content, temperature and liquid-ice phase partitioning, and a selection of effective medium approximation available for ice-air and water-ice-air mixtures. Since SB2M does not have a prognostic melted fraction, B-PRO uses melting parameterization for treatment of melting hydrometeors. Table 3 summarizes the parameters used to estimate the scattering properties of the modeled hydrometeors in the forward operator. The diameter size distribution $f(D_x)$ is calculated for all hydrometeors based on the estimated parameters of the Gamma distribution $A$ and $\lambda$ (Eq. 1) using the shape parameter (Table 1) and model outputs of $q_x$ and $N_x$. For rain below clouds ($q_c = 0$), the shape parameter is diagnosed from of $q_r$ and $N_r$, using the parameterization of the shape of the raindrop size distribution as a function of the mean volume diameter (Seifert, 2008). More details about the B-PRO is also available from (Shrestha et al., 2022).

Since T-matrix computations are computationally very expensive in the absence of look-up tables, B-PRO simulations are performed only for a cropped model domain (180x180x80 grid points) and for limited time periods. We also decomposed the model grid area into smaller sub-domains (20x20x80 grid points), such that B-PRO can be run in parallel in order to further speed-up the T-matrix computations.

## 3 Polarimetric Radar Observations

The observed polarimetric radar variables used in this study are based on the twin research X-band Doppler radars located in Bonn and Jülich (BoxPol and JuXPol; Diederich et al. 2015a, b), which operate at a frequency of 9.3 GHz with a radial resolution of 100-150 m and a scan period of 5 minutes. Both X-band Doppler radars produce volume scans consisting of a series of Plan Position Indicators (PPIs) measured at ten different elevations, mostly between 0.5 ° and 30 °, followed by a vertical cross-section (RHI - range height indicator) in a specific direction and a vertically pointing scan. The use of these multiple PPI sweeps became more popular in recent years in order to get a 3D picture of surrounding hydrometeors and microphysical

processes. These PPIs can be exploited for improved process understanding, model evaluation and data assimilation. And, such volume scans also enable us to construct vertical cross-sections of convective systems.

$Z_H$ was calibrated by comparison with observations of the Dual-frequency Precipitation Radar (DPR) onboard the Global Precipitation Mission (GPM) Core Observatory satellite. To this goal, both observations are first brought to the same observational volumes, then the melting layer is identified and excluded from the calculation of the median. The calibration based on GPM DPR (Ku-band) is consistent with results obtained with the methodology described in Diederich et al. (2015a). Furthermore, the calibration technique selects only stratiform events where a bright band is visible, and only reflectivities between

$10\,dBZ$ and $36\,dBZ$ are taken into account, to avoid strong effects of attenuation. Successful calibrations of ground-based radars with satellite-based radars have been also been done in several previous studies (Schwaller and Morris, 2011; Protat et al., 2011; Warren et al., 2018; Crisologo et al., 2018; Louf et al., 2019).

The $Z_{DR}$ calibration uses vertical scans where near-zero $Z_{DR}$ are expected. Values with $\rho_{hv} < 0.9$ are filtered out to avoid impacts of non-meteorological scatterers, and $Z_H > 30\,dBZ$ are ignored to keep only stratiform events. The melting layer and

the near-radar gates (first $600\,m$) are also removed to reduce noise and the offset calculated as the median of the remaining values (Williams et al., 2013; Ryzhkov and Zrnic, 2019). Futher adjustments are made for both $Z_H$ and $Z_{DR}$ based on a comparison between BoXPol and JuXPol. The radar calibration varies with time; see table in A1 for observed offsets for the different events.

Besides radar miscalibration and partial beam blockage, the polarimetric variables $Z_H$ and $Z_{DR}$ are affected by (differential)

attenuation, especially at smaller wavelengths (C band and X band), and their correction especially in deep convective, hail-bearing cells gives rise to additional uncertainties( Snyder et al. 2010). Corrections for attenuation and differential attenuation especially due to hail follows the algorithm from Ryzhkov et al. (2013). The algorithm first identifies radial segments with potential hail along the beam via $Z_H > 50\,dBZ$. For these segments, the coefficient for attenuation is calculated via the ZPHI method from Testud et al. (2000). Differential attenuation due to the presence of hail is calculated by comparing the observed

$Z_{DR}$ behind the hotspot with an expected value based on $Z_H$ (at values between 20 and 30 dBZ) to ensure light rain, Eq. 11 in Ryzhkov et al. (2013) and use the difference to calculate the value of the differential attenuation coefficient in the hail core. For other segments, the standard linear relationships between attenuation and differential attenuation and differential phase ($\phi_{DP}$) are used with standard coefficients for X band from Ryzhkov and Zrnic (2019) ($\alpha = 0.28$ and $\beta = 0.03$). These coefficients are not used for the hail inflicted segments for which we do not know the actual attenuation and differential attenuation—the

above method only provides estimates of attenuation-corrected $Z_H$ and $Z_{DR}$.

In contrast, $K_{DP}$ is not affected by miscalibration and attenuation. However, the total differential phase shift is a combination of backscatter differential phase ($\delta$) and propagation differential phase ($\varphi_{DP}$); thus the subtraction of the former from the total differential phase shift ($\Phi_{DP}$) is required before computing $K_{DP}$. This is particularly important when hydrometeor sizes are in the range of or larger than the radar wavelength; these so-called resonance effects are most pronounced at C band but also

significant at X band (Trömel et al., 2013). Once the contribution of ($\delta$) is removed, $K_{DP}$ is estimated by calculating the range derivative of $\varphi_{DP}$. We acknowledge this uncertainty in the estimates of attenuation corrected radar observations, and identifying the contribution of ($\delta$) affects, which can affect the $K_{DP}$ estimates.

Based on the time and location of the storm from the radar, PPI measured at different elevation for each case are used, giving insights of the measurement of convective systems at different heights ( 1 km, near melting layer and 2-3 km above melting layer). We also further interpolated the polarimetric radar data from the native polar coordinates to cartesian coordinates at 500 m horizontal and vertical resolution using a Cressman analysis with a radius of influence of 2 km in the horizontal and 1 km in the vertical. While, the data in native polar co-ordinates is used for investigating polarimetric signatures, the gridded data allows for easy comparisions with their model-simulated equivalents. Ground clutter and non-meteorological scatterers are known for having significantly decreased values of $\rho_{hv}$ compared to precipitation (Zrnic and Ryzhkov, 1999; Schuur et al., 2003). Therefore, a threshold of 0.8 in $\rho_{hv}$ was imposed in the gridded data to ensure that clutter is filtered out without removing useful meteorological information.

Besides the observations from the X-band radars, the RADOLAN (Radar Online Adjustment; Ramsauer et al., 2018; Kreklow et al., 2020) data from the German national meteorological service (DWD, Deutscher Wetterdienst) is also used for evaluating the modeled precipitation. RADOLAN is a gauge adjusted precipitation product based on DWD's C-band weather radars available at hourly frequency in a spatial resolution of 1 km.

## 4 Experiment Setup

The model evaluation with polarimetric radar data is conducted for three cases of summertime convective storm events producing hail, heavy precipitation and severe winds. Figure 1 shows the synoptic conditions for the three cases; shown are the surface pressure reduced to mean sea level and pseudo-equivalent potential temperature based on GFS analysis at 1200 UTC. Additional synoptic plots are also directly available from http://www1.wetter3.de. The first case (5 July 2015) is a northeastward propagating deep convective hail-bearing storm crossing Bonn. The storm was associated with a low-pressure system west of Ireland with an occluded front crossing Norway and the cold front extending over the western part of middle Europe producing pre-frontal convergence zones over western Germany, where a moisture tongue ahead of the cold front produced instability and drew warm moist air mass from the south (Fig. 1 a). Scattered notheasterly propagating storms were prevalent throughout the day, with an isolated deep convective storm passing directly over the Bonn radar from 1500 to 1600 UTC. Acccording to the European Severe Weather Database (ESWD), large hail (2 - 5 cm in diameter) was observed over the Bonn region, including damaging lightning further north, and heavy precipitation with severe wind (further north-east).

The second case (13 May 2016) is chracterized by scattered convective storms over Rheinland Pfalz, Germany, associated with a low pressure system over the Norwegian sea with an occluded front over northern and a cold front over southern Germany (Fig. 1 b). The southward propagating cold front provided the necessary lift to release the potential instability associated with a warm moist air mass below 700 hPa over the region between the occlusion and the cold front. The ESWD reported heavy rainfall over the Frankfurt area resulting in flooding and damage to property.

The third case (6 July 2017) consists of deep convective clouds propagating eastwards over Bonn. On that day, a warm front over central Germany separated a relatively cool northern, from a warm southern Germany (Fig. 1 c). The additional northward push of the warm front produced the necessary lift to release the potential instability associated with the warm and

moist southerly air mass. The ESWD reported scattered severe wind around the Bonn region and heavy precipitation south of Mainz including large hail.

## 4.1 Model Domain

The experiment is setup over the Bonn Radar domain (Shrestha, 2021a) - a temperate region in the northwestern part of Germany bordering with the Netherlands, Luxemburg, Belgium, and France (Fig. 2a). The region has a quite heterogeneous land cover and comprises extensive emissions by point (e.g., oil refineries, photochemical industries) and area sources (e.g., extensive urban and rural areas, road transport, extensive agriculture, railways) (Kulmala et al., 2011; Kuenen et al., 2014). The twin polarimetric X-Band research radars in Bonn (BoxPol) and Jülich (JuxPol) and the overlapping measurements from four polarimetric C-Band radars of the German Weather Service (Deutscher Wetterdienst, DWD) make the region probably the best radar-monitored area in Germany. The model domain covers approximately $333x333\,km^2$ area with a horizontal grid resolution of $1.132\,km$. Eighty level are used in the vertical with a near-surface-layer depth of $20\,m$ for the atmospheric model. For the hydrological model, 30 vertical levels with 10 stretched layers in the root zone ($2–100\,cm$) and 20 constant levels ($135$ cm) below is used, extending down to $30\,m$ below the surface.

The land cover type and associated phenology is based on the Moderate Resolution Imaging Spectroradiometer (MODIS) remote sensing products (Friedl et al., 2010; Myneni et al., 2015). The Rhein massif intersected by the Middle Rhein valley dominates its topography, and the land-cover consists of forested areas (58%), agricultural land (23%), urban areas (12%) and grasslands (7%).

## 4.2 Simulations

Ensemble simulations with 20 members for three case studies are used to quantify the meteorological uncertainty in simulated precipitation and polarimetric variables. The hourly model output from the 20 ensemble members of the COSMO-DE Ensemble Prediction System (EPS; Gebhardt et al., 2011; Peralta et al., 2012) provided by DWD is used for the model runs in this study. The COSMO-DE is a high resolution ($2.8\,km$) configuration of the COSMO model encompassing the entire extent of Germany. The 20 ensemble members of COSMO-DE EPS can be divided into 4 subsets of 5 members each. The 4 subsets represent different global models: the Integrated Forecast System of ECMWF (IFS; ECMWF, 2003), the global model of DWD (GME; Majewski et al., 2002), the Global Forecast System of NCEP (GFS; Center, 2003) and the Unified Model of the UK Met Office (UM; Staniforth et al., 2006), used to vary the boundary conditions of the COSMO-DE. Each subset of the 5 members is then perturbed by varying a set of parameters that control the physics parameterization of the COSMO model. The general statistics of the EPS are always stratified according to four global models when used for IC/BC perturbations of COSMO-DE; i.e. the five members having the same global model are more similar to each other (personal communication: G. Christoph, DWD). Since January 2015, the ICON (ICOsahedral Non-hydrostatic Zängl et al., 2015) modelling framework was used instead of the global numerical weather prediction model GME (Majewski et al., 2002). Also, the EPS system was switched to BCs based on ICON-EU-EPS and IC perturbations generated by a Local Ensemble Kalman Filter from March 2017 onwards.

The initial soil-vegetation states are obtained from spinups using offline hydrological model runs over the same domain (Shrestha, 2021b). In all runs, a coupling frequency of 90 s is used between the atmospheric and hydrological components, which have a time steps of 10 s and 90 s, respectively. The models are integrated over diurnal scale starting at mid-night. The atmospheric model output is generated at 5 min intervals, while the hydrological model output is generated at hourly intervals. For the third case, the internal variability in the ensemble members was relatively high in terms of the spatio-temporal distribution of convective storms (probably associated with the switching of the ensemble generator in 2017); thus the output was generated at 15 min intervals over a longer model period in order to allow for a fair comparison with observations and to maintain the same load for synthetic polarimetric processing and data storage.

The ensemble simulation per event required an average of 54 core-hours using 456 compute cores on the JUWELS (Jülich Wizard for European Leadership Science) machine at Jülich Supercomputing Center (JSC). Approximately 540 GB of data were produced per event. For polarimetric variables, only 3 hourly data containing 37 time snapshots were processed for each simulation on a local linux cluster (CLUMA2), amounting to 220 GB per event.

## 5  Results

### 5.1  Accumulated Precipitation

First, we examine the model simulated ensemble precipitation with the RADOLAN data. Figure 3 shows the spatial pattern and frequency distribution of the modeled and observed accumulated precipitation over the Bonn Radar domain for the three case studies. Overall, the spatial pattern of ensemble averaged accumulated precipitation resemble the RADOLAN estimates, but the frequency distribution produced by the ensemble members underestimate high precipitation. For the first case (Fig. 3 a), the model simulated accumulated precipitation is stratified according to four global models used for IC/BC. The members using GME data produce average accumulated precipitation and a frequency distribution for average accumulated precipitation (< 30 mm) closest to RADOLAN. The model does, however, underestimate average accumulated precipitation (> 30 mm) for all ensemble members as also visible in the spatial pattern of the ensemble averaged accumulated precipitation. While the large-scale extent of the precipitating area is comparable between model and RADOLAN, the precipitation amount especially in the northeastern domain is underestimated. For the second case (Figure 3b), all ensemble members underestimate the average accumulated precipitation compared to RADOLAN; also its frequency distribution for high precipitation is weaker compared to the first case. All ensemble members for second case, underestimates average accumulated precipitation (> 10 mm). For the third case (Fig. 3 c), the model misses the precipitation observed over the western part of the domain for all ensemble members except of one, and the simulated frequency distribution of accumulated precipitation exhibits a larger spread. This could be attributed to the switch in the ensemble generator for large scale atmospheric forcing data.

## 5.2 Polarimetric Signatures

For a given precipitation type, polarimetric variables are expected to cluster in a specific region of the multi-dimensional space (Zrnic and Ryzhkov, 1999). Thus as one evaluation method, we compare the respective clustering between simulations and observations for similar stages of convection, which we identify via the Convective Area Fraction (CAF: area fraction of a storm with radar reflectivity >40 dBZ at 2 km height above ground level (hereafter a.g.l.); Fig. 4) and by a qualitative exploratory analysis of the model ensembles and the observed storm evolution. The total area of the storm for CAF estimate, includes the grid points of the storm with radar reflectivity >0 dBZ at 2 km height a.g.l. The time extent of the CAF evolution was chosen such that the storm is within the domain. However, due to variability in the ensemble members, some members are affected as part of the storm approaches the boundary in the last 30 minutes of CAF evolution for Case 1 and 2. For Case 3, due to extended sampling time used, the CAF is also partly impacted by the storm moving off the grid for the synthetic data. For the first case, the observed storm CAF decreases while approaching the radar and increases again while moving away from the radar. Especially, the ensemble members initiated and forced with GME model (relatively dark lines) show a similar behaviour but underestimate CAF compared to observations. For the second case, CAF gradually increases for all ensemble members and remains quasi-steady after 1100 UTC. However, all ensemble members underestimate CAF in the earlier phase of the storm (before 1100 UTC) compared to observations. For the third case, the simulated CAFs of the model ensembles have a wider spread, probably caused by a switch in the way the ensemble is generated from March 2017 onwards. While few ensemble members simulate the storm much earlier than observed (relatively dark lines), the CAF of one ensemble member, better matches the observations and exhibits also a storm evolution (dark line) quite similar to the observations.

The comparison of model with observation is always challenging, due to mismatches of the simulated and observed storm evolution in space and time (also shown by the variability in the CAF evolution). So, besides exploring the time series of CAF, we also conducted a qualitative exploratory analysis (using synthetic polarimetric variables at lower levels ( 1000 m a.g.l.), mid levels (near melting layer), and upper levels (2.5 km above melting layer) to find the simulated convective storm among the ensemble members that was closest in time and location compared to the polarimetric observations. Importantly, a qualitative exploratory analysis of the PPIs (at different elevations) and reconstructed RHIs of observed polarimetric variables were also conducted to identify prominent polarimetric signatures. Based on the above two analyses, we identified the ensemble members, time-snapshot (identified by square markers in Fig. 4) and time intervals (solid lines bounded by vertical bars in Fig. 4) for the comparison of the polarimetric features and statistical distribution of polarimetric variables between observations and simulations respectively.

Here, we have to note that, both synthetic and observed radar variables are affected by errors in forward operator and calibration/attenuation corrections respectively. We acknowledge this limitation in the study, and concentrate more on the prominent patterns and not so much on the actual magnitudes of the polarimetric variables.

### 5.2.1 Case One

Fig. 5 a shows the Plan Position Indicator (PPI) plots of $Z_H$, $Z_{DR}$, $K_{DP}$ and $\rho_{hv}$ at 8.2 degree elevation observed by BoXPol at
1530 UTC for the first case. The storm is characterized by high reflectivity (>50 dBZ) and differential reflectivity (> 2 dB) near
the melting layer. An arc-like feature of high $Z_{DR}$ follows the leading eastern edge of the storm just below the melting layer
with concurrent lower $Z_H$ values suggesting hydrometeor size sorting associated with storm inflow (Kumjian and Ryzhkov,
2012; Dawson et al., 2014; Suzuki et al., 2017). Fig. 5 b shows a cross-section of storm based on the gridded radar data. Its
convective part between -20 and 5 km relative to BoXPol exhibits a notable polarimetric feature - $Z_{DR}$ columns, anchored to
lower levels and extending up to 6 km altitude associated with two strong updraft zones. They are associated with the presence
of supercooled rain drop, water-coated hail growing in wet growth regime and frozen rain drops aloft, and their different
extensions suggest different updraft intensities (Kumjian and Ryzhkov, 2008; Kumjian et al., 2014; Snyder et al., 2015).
$K_{DP}$ columns (Ryzhkov and Zrnic, 2019; Snyder et al., 2017b) co-located with the $Z_{DR}$ columns are another prominent
polarimetric feature with slight inward offsets that are considered additional signs for updraft locations and presence of liquid
water associated with either supercooled raindrops or water-coated hail ( van Lier-Walqui et al. 2016). The low (<0.7) cross-
correlation coefficient ($\rho_{hv}$) near the inflow region and the even lower $\rho_{hv}$ (<0.92) along the strong convective core associated
with high reflectivity probably indicates hail. The dominance of near-zero $Z_{DR}$ and reflectivity values between 20 and 25 dBZ
above the melting layer in the downdraft region suggest the dominance of snow (Yuter and Houze Jr, 1995). The low $\rho_{hv}$ in the
northern region at higher levels associated with relatively high $Z_{DR}$ and moderate $K_{DP}$, are probably caused by horizontally
oriented ice crystals.

As discussed in Sect. 5.1, the ensemble members initiated using GME data have similar storm evolutions as observed. So,
only these ensemble members are used here for the polarimetric comparisons. Fig. 6 shows the synthetic polarimetric moments
at lower levels up to the melting layer and cross-sections of polarimetric variables and simulated hydrometeors at 1455 UTC
for one of the ensemble members (Fig. 4 a—dark solid line). At lower levels (1000 m a.g.l.), the southeastern flank of the
storm has - as expected near the core of the storm - relatively high $Z_H$ and $Z_{DR}$ (also associated with relatively low $\rho_{hv}$) with
lower magnitudes on the northwestern side. $K_{DP}$ has generally low magnitudes while $\rho_{hv}$ is generally high. Near the melting
level (4000 m a.g.l.), $K_{DP}$ present much lower magnitudes but a ring like feature in $Z_{DR}$ with relatively low $\rho_{hv}$ is visible
in the convective core, which is a typical polarimetric feature found for supercell storms (Kumjian and Ryzhkov, 2008). This
enhanced $Z_{DR}$ found in observations are hypothesized to be contributed by ice-phase hydrometeors upon melting or accretion
of liquid water (Ryzhkov and Zrnic, 2019). Here, the synthetic elevated $Z_{DR}$ is primarily contributed by the the elevated
perturbation of warm temperature in the convective core and the melting of ice-phase hydrometeor, which is parameterized in
the FO.

In all ensemble members, the storm is aligned in the northeast direction and has a strong updraft region in the southeasten
edge characterized by a bounded weak echo region (BWER, see Fig. 6 c). The convective storm top extends up to 15 km
height with $Z_H$ between 30 and 40 dBZ (which is relatively lower than the observed $Z_H$) co-located with the simulated hail
shaft and updraft (Fig. 6 d). The model also exhibits a narrow $Z_{DR}$ column like feature extending up to 6 km altitude in the

convective core. However, the simulated $Z_{DR}$ column is relatively smaller in width and magnitude (value) compared to the observations. The synthetic $Z_{DR}$ column signature is a result of supercooled raindrops with mean diameter size of 1.3-1.7 mm. The model also simulates high $K_{DP}(> 1$ deg/km$)$ along the top of the convective storm part, but no $K_{DP}$ columns are present

adjacent to the updraft region above the melting layer as seen in the observations. Although, the simulated $\rho_{hv}$ is higher than observed, slight decrease can be observed in the updraft region with high $Z_H$ associated with hail, in the $Z_{DR}$ column and below the melting layer. In the updraft region, the modelled vertical velocity above 8 km reaches 40 m/s, dominated mostly by super-cooled raindrops around 6-9 km (see Fig. 6 d), which is an important source for hail growth. The strong updraft also generates a warm anomaly above the melting layer (see the $0°$ isotherm) in the simulations, below which rain is also formed by

melting of graupel and hail. Graupel dominates the frozen hydrometeor categories above the melting layer peaking at the top of the updraft region. Ice crystals are located mostly above 8 km height, and the self-collection of these ice particles leads to the formation of snow which further grows in size via aggregation. Hail is present in low concentration in the convective core, but contributes dominantly in the polarimetric signals in terms of high reflectivity, $Z_{DR}$ (especially below the melting layer) and lower $\rho_{hv}$.

**5.2.2   Case Two**

Fig. 7 shows the PPIs of $Z_H$, $Z_{DR}$, $K_{DP}$ and $\rho_{hv}$ at 1.0 degree elevation from BoXPol at 1030 UTC for the second case. We find moderate reflectivities (35 - 40 dBZ) and high $Z_{DR}$ (>2 dB) at around 1 km. According to the cross-section of storm based on gridded polarimetric radar data (Fig. 7 b), the storm has a wide $Z_{DR}$ column like feature anchored to the lower levels and extending up to 5 km. At this location, below the melting layer (approx. 2.5 km), $Z_{DR}$ is >2 dB while reflectivity is weak,

which suggests size-sorting of rain drops. A large portion of the storm exhibits very low or negative $Z_{DR}$ above the melting layer, possibly indicating vertically oriented or conical graupel (Bringi et al., 2017). While other studies also have shown the presence of low and negative $Z_{DR}$ above melting layer for convective storms ( Suzuki et al. 2017; Hubbert et al. 2018), it is possible that for these convective cases, attenuation correction even with the advanced methods as we used here may at least partially contribute to negative $Z_{DR}$.

Figure 8a,b shows the synthetic polarimetric moments up to near the melting layer and cross-sections of polarimetric variables and simulated hydrometeors at 1050 UTC for one of the ensemble members (see Fig. 4 b—thick solid line). The southwards propagating storm is oriented in north-south direction. Regions with moderate to high reflectivities in the lower levels (1000 m a.g.l.) coincide with moderate to high $Z_{DR}$, $K_{DP}$ and lower $\rho_{hv}$ suggesting heavy rain or rain/hail mixtures. Just above the melting level (3000 km a.g.l.), $Z_{DR}$ and $K_{DP}$ are much lower except on the western storm edges, where slighly enhanced

$Z_{DR}$ and $K_{DP}$ features are found. According to the cross-section (Fig. 8 c), moderate reflectivities (30-50 dBZ) comparable to the observations, reach up to 6 km height while the storm top height extends up to 9 km. The model does not capture a distinct $Z_{DR}$ column but simulates narrow region with enhanced $Z_{DR}$ and lower $\rho_{hv}$ above the melting layer, extending up to 7 km (Fig. 8d). The simulated enhanced $Z_{DR}$ is due to the presence of supercooled raindrops with mean diameter size of 0.7-0.9 mm. A grid-scale enhanced $K_{DP}$ extending up to 4 km above the melting layer is also visible but $K_{DP}$ generally, remains very low

here except for some region near the storm top, which is also visible in the observations.

Based on the modeled hydrometeors, Fig. 8 d indicates presence of super-cooled raindrops above the melting layer connected with updraft regions (5 m/s maximum vertical velocity at the left and right edges of the storm). However, the smaller size of raindrops (< 1 mm) are not sufficient to create strong $Z_{DR}$ magnitudes as observed in the $Z_{DR}$ columns. The vertical velocity in the storm center is around 1 m/s and not included in the contour plot. The frozen hydrometeors are again dominated by graupel with high concentrations in the strong updraft region. Hail is present in low concentrations, adjacent to the updraft regions reaching down to the surface. Above 6 km height, some cloud ice exists while this region is mostly dominated by snow.

### 5.2.3 Case Three

Fig. 9 shows the PPIs of $Z_H$,$Z_{DR}$, $K_{DP}$ and $\rho_{hv}$ at 8.2 degree elevation from BoXPol at 1400 UTC. The storm is characterized by reflectivities > 50 dBZ and $Z_{DR}$ >2 dB near the melting layer. Its convective region (reflectivities´ > 50 dBZ) extends up to 12 km height and the corresponding lower $\rho_{hv}$ indicate presence of hail (Figure 7b). The convective core has also relatively high $K_{DP}$ values extending up to the storm top and including a wide $Z_{DR}$ column up to 5 km height. Both indicate lofting and growth of large rain drops by updrafts, which are also important for hail formation. This case also shows low to negative $Z_{DR}$ values above the melting layer, which could also be partially contributed by limitations on the attenuation correction algorithm.

Fig. 10 shows the plan view of synthetic polarimetric variables (at lower levels and near melting layer) and a cross-section of them including hydrometeors at 1530 UTC simulated by one of the ensemble members (see Fig. 4 c—thick solid line). The eastward propagating storm is oriented from west to east and at lower levels characterized by a wide core of moderate reflectivity (40-50 dBZ) and high $K_{DP}$, $Z_{DR}$ >2 dB along the edges, and low $\rho_{hv}$ produced by heavy rain and rain/hail mixtures. Near the melting level (4000 m a.g.l.), variable $Z_{DR}$ and $Z_H$ features are found near the southeastern edge—characteristics of rain drop size-sorting. Overall, $Z_{DR}$ and $K_{DP}$ are low throughout the storm. According to the cross-section (Fig. 10 c), the storm extends up to 12 km with moderate reflectivities (30-50 dBZ). While, $Z_H$ at lower levels is comparable to observations, the relatively high $Z_H$ seen in the observations extending up to upper levels is underestimated by the model. The model also simulates a narrow $Z_{DR}$ column extending up to 5 km adjacent to the updraft region and relatively comparable to observation. The $Z_{DR}$ column signature is a consequence of supercooled raindrops with mean diameter size of 1.7-1.9 mm. The convective core also has relatively higher $Z_{DR}$ than the background, extending up to 12 km height. The model also simulates high $K_{DP}$ along this convective part of the storm. The simulated $\rho_{hv}$ is again generally high with slight decrease in the convective core and below the melting layer, an indication of hail, together with the high $Z_H$. Similar features of $Z_{DR}$, $K_{DP}$ and $\rho_{hv}$ is also seen in the observed convective core.

The vertical velocity reaches to 10 m/s from 6-11 km in the updraft region where a low concentration of super-cooled rain drops is found up to 8 km (Fig. 10 d). Graupel again dominates the frozen hydrometeor categories above the melting layer, while snow further extends downwards up to 6 km height. Compared to the other two cases the simulated hail concentration is relatively higher and contributes dominantly to the polarimetric signatures.

### 5.3 Frequency distribution of polarimetric variables

Because mismatches between space and time scales of synthetic polarimetric moments compared to observations are present, ensemble properties of the convective event are also monitored. For this purpose, the ensemble simulations are compared to the observations for similar storm evolution stages using contoured frequency altitude diagrams (CFADs; Yuter and Houze Jr 1995) using the same extents and bin widths for observations and simulations.

#### 5.3.1 Case One

We use the observations from 1445 to 1530 UTC, which encompasses the convective stage of the storm before it passes over the BoXPol. The CFADs from the X-band radar (Fig. 11 a) show a unimodal distribution of $Z_{\mathrm{H}}$ which gradually narrows above the melting layer (around 4 km). The peak in the frequency distribution occurs around 20-25 dBZ with maximum reflectivities well above 50 dBZ. The $Z_{\mathrm{DR}}$ also exhibits a unimodal distribution which further peaks (or narrows) above the melting layer with the mode around 0.25 dB, similar to the values reported by (Yuter and Houze Jr, 1995) for convective storms. The distribution broadens and shifts to values up to 4 dB below the melting layer peaking at around 1 dB near the surface. $K_{\mathrm{DP}}$ exhibits a unimodal distribution throughout the vertical extent of storm with peak values around 0.1 deg/km. The distribution also broadens weakly from 7 km height downwards. $\rho_{\mathrm{hv}}$ has a quite broader distribution peaking around 0.98 below 11 km height and shifting to 0.87 near the storm top.

The CFADs from the model ensemble were generated using five members from 1445 to 1530 UTC (Fig. 4 a—soild lines) which best matched the observed storm macrophysical features. The $Z_{\mathrm{H}}$ distribution with maximum reflectivities generally below 50 dBZ peaks around 28 dBZ from 6 to 10 km, but shifts towards 15-20 dBZ at lower levels, which were found to be associated with grid cells with very low concentration of hydrometeors broadening the distribution, compared to observations. $Z_{\mathrm{DR}}$ again exhibits a narrow unimodal distribution above melting layer peaking around 0.1 dB, which broadens below the melting layer with an additional peak at 2.6 dB. Unlike the unimodal CFADs from observations, the CFADs from the model ensemble produce bimodal peaks below the melting layer. $K_{\mathrm{DP}}$ shows a very narrow unimodal distribution compared to the observations with peak values around 0.1 deg/km. For the given range (0.7-1.0) of $\rho_{\mathrm{hv}}$, the frequency distribution appears to be poorly simulated by the model.

#### 5.3.2 Case Two

CFADs are generated during the convective period of the storm from 1010 to 1055 UTC. The $Z_{\mathrm{H}}$ observations (Fig. 12 a) show a unimodal distribution peaking around 25 dBZ and gradually narrowing above the melting layer ( 3 km) with maximum reflectivities > 45 dBZ. $Z_{\mathrm{DR}}$ also exhibits a unimodal distribution peaking above the melting layer at around -0.12 dB but broadening and shifting to higher values with peaks around 0.4 dB near the surface and maxima > 2 dB below the melting layer. Compared to case one, a leftward shift can be observed for the $Z_{\mathrm{DR}}$ distribution, which is primarily caused by domination of low to negative $Z_{\mathrm{DR}}$ above the melting layer. But, similar to the first case, $K_{\mathrm{DP}}$ has a unimodal distribution throughout the

storm with peak values around 0.1 deg/km with a very weak broadening downwards and below the melting layer. $\rho_{\text{hv}}$ exhibits again a broader distribution peaking around 0.97 (below 7 km height) and shifting to 0.85 near the storm top.

The CFADs from the model ensemble were generated from 5 members from 1030 to 1115 UTC (see Fig. 4 b—soild lines). The CFADs for $Z_{\text{H}}$ have a broader distribution compared to observation with maxima generally below 45 dBZ; the distribution peaks around 28 dBZ near the melting layer (around 3 km) and gradually shifts towards 10 dBZ near the storm top (around 8 km) and towards 32 dBZ below the melting layer. $Z_{\text{DR}}$ has a narrow unimodal distribution above the melting layer peaking around 0.12 dB. The CFAD broadens below the melting layer with an additional peak at 2.5 dB. Again, the model CFADs produce

bimodal peaks compared to unimodal distribution for observations. Additionally, no leftward shift in the $Z_{\text{DR}}$ distribution is observed for model ensembles as seen in observations compared to case one. $K_{\text{DP}}$ also shows a very narrow unimodal distribution compared to the observations, peaking around 0.12 deg/km. The distribution weakly broadens below the melting layer and at upper levels. For the given range (0.7-1.0) of $\rho_{\text{hv}}$, the frequency distribution again appears to be poorly simulated by the model.

### 5.3.3  Case Three

CFADs are generated from 1330 to 1415 UTC. The observed unimodal $Z_{\text{H}}$ distribution (Fig. 13 a) has maxima > 50 dBZ and a peak around 25 dBZ which gradually narrows above the melting layer around 4 km and shifts to smaller values peaking around 17 dBZ upwards above 9 km. $Z_{\text{DR}}$ also exhibits again a unimodal distribution above the melting layer with peak around -0.12 dB. The distribution broadens and shifts to larger values below the melting layer peaking around 0.4 dB near the surface

with maxima > 2 dB. The $Z_{\text{DR}}$ distribution is similar to case two. $K_{\text{DP}}$ again exhibits a unimodal distribution with peak values around 0.1 deg/km and weakly broadens below the melting layer. Again, $\rho_{\text{hv}}$ has a broad distribution peaking around 0.98 (below 8 km height) but shifting towards 0.83 at the storm top.

The CFADs from the model ensemble were generated using only 1 ensemble member from 1500 to 1545 UTC (see Fig. 4c—solid line) due to strong variability among the ensemble members. The CFADs for horizontal reflectivity have maxima

below 50 dBz and again exhibit a broader distribution compared to observations, peaking around 8 and 38 dBZ near the melting layer (around 4 km) producing two peaks, and shift towards 10 dBZ near the storm top (around 10 km) and towards 42 dBZ near the surface. $Z_{\text{DR}}$ has a narrow unimodal distribution above the melting layer with a peak around 0.1 dB and broadens below the melting layer with an additional peak at 1.5 dB. The model again produces bimodal peaks below the melting layer and additionally do not show any leftward shift in the $Z_{\text{DR}}$ distribution as seen between observations for case three and one.

$K_{\text{DP}}$ also shows again a very narrow unimodal distribution with peak values around 0.1 deg/km which broadens both below the melting layer and at upper levels. For $\rho_{\text{hv}}$, the frequency distribution again appears to be poorly simulated by the model.

## 6  Discussion

The variability in the lateral boundary conditions for the ensemble members was found to generate probabilistic forecast in the accumulated precipitation and convective area fraction (Gebhardt et al., 2011). The lateral boundary conditions affect

the simulated cloud microphysical and macrophysical processes and hence the synthetic polarimetric variables. However, the magnitude of this influence varies between the three studied cases. Particularly, the switch in the ensemble generation for the third case produced a much stronger variability in the spatio-temporal structure of the simulated storm. The CAFs from observations and model simulations indicate that the initial intensity of storms are underestimated by the model, which partly explains the underestimation of high precipitation for all ensemble members. In simulations by Noppel et al. (2010) for a

hail storm over southwestern Germany using the same atmospheric model COSMO with the two moment microphysics, the continental CN concentration $(1700\,cm^{-3})$ led to a weaker storm and less surface precipitation compared to maritime CN concentrations $(100\,cm^{-3})$. However, their additional sensitivity study by varying the fixed parameters in Eq. 1 for cloud hydrometeors in order to produce a narrow distribution led to a different conclusion, indicating a missing feedback between the CN concentration and the shape parameters of the cloud droplet size distribution (which are both fixed in the model). This

mechanism could also be partly contributing to the weaker initial intensity of the storms presented in this study.

The polarimetric radar observations for the three case studies of summertime convective storms exhibits a prominent $Z_{DR}$ and $K_{DP}$ columns indicating convective updrafts. In general, the synthetic radar data shows that the model is able to capture the prominent polarimetric signature of the observed convective storms like the $Z_{DR}$ columns, besides other additional signatures (e.g., size sorting and the ring like feature of $Z_{DR}$ with relatively lower $\rho_{hv}$ typically observed in supercells). However, the

distinct $K_{DP}$ columns observed especially in Case one and three are not captured by the model. Further, a relatively enhanced $Z_{DR}$ compared to the background is also captured by the model in the convective core for all case studies, which is also present in the observations. While the synthetic $Z_{DR}$ column for case three was close in magnitude to the observed radar data, the model was found to generally underestimate the width and the magnitude (value) of the $Z_{DR}$ column and its anchorage to the ground, compared to observations. The synthetic $Z_{DR}$ column signature is a result of the supercooled raindrops only.

The missing treatment of freezing raindrops (which do require an additional hydrometeor class) could also be contributing to deficiency in the polarimetric signature (Kumjian et al., 2014). And, to a certain extent, the absence of polarimetric signature contribution from wet growth of hail, which is not parameterized in the FO could additionally be contributing to the deficiency in the shape and magnitude of the synthetic $Z_{DR}$ column. Besides, the mean diameter size of the raindrops strongly control the magnitude of polarimetric signature. A reason for relatively small mean diameter size of supercooled raindrops could be

due to high CN concentrations and the missing feedback between the CN concentration and shape parameters of cloud drop size distribution (Noppel et al., 2010). A sensitivity study with low CN concentrations for case one infact produced high hail concentration, which increased the CAF, $Z_{DR}$ and $Z_{H}$ magnitudes of the storm (Trömel et al., 2021).

Below the melting layer in the downdraft regions, where the melting of graupel and hail are the main source of rain water and produce high $Z_{DR}$, simulations generally replicate the observations. Above the melting layer, the partitioning of the ice water

content in the model is generally dominated by graupel for all case studies. The dominance of graupel has also been reported in previous modelling studies (Pfeifer et al., 2008; Tao et al., 2011; Lang et al., 2011; Shrestha, 2011; Shrestha et al., 2015). E.g. similar finding to this study was also reported earlier by Pfeifer et al. (2008) for a squall line over Germany, where they showed that the simulated ice-phase hydrometeors were mostly dominated by graupel while the observation showed the dominance of snow. In this study also, case one with near zero $Z_{DR}$ and reflectivities between 20-25 dBZ, indicate domination of snow in the

downdraft region. However, low to negative $Z_{DR}$ above the melting layer for case two and three possibly indicate domination of graupel, but we cannot be completely certain as it might be partially affected by the attenuation correction algorithm as discussed above.

The statistical properties of the observed polarimetric variabiables exhibit similar patterns for all three case studies in terms of CFADs. In general, the $Z_H$-CFADs from the observations exhibit narrow unimodal distributions peaking around 20-25 dBZ,
but differ in maximum reflectivities (>50 dBZ for case one and three, >45 dbz for case two). Similarly, the observed CFADs for $Z_{DR}$ also show unimodal distribution above the melting layer, which gradually shifts towards higher value near the surface for all three cases. While the pattern of $Z_{DR}$ CFADs is similar for observations in all cases, the location of the peaks above the melting layer differ between case one (0.25 dB) and other two cases (-0.12 dB). This difference in the peak of the observed $Z_{DR}$ distribution could also point towards the possible difference in partitioning of ice water content above the melting layer
as well as partial effect of attenuation correction algorithm. The $K_{DP}$-CFADs exhibit a narrow unimodal distribution for all case studies, while $\rho_{hv}$ CFADs exhibit a broader distribution with peak around 0.97-0.98, which shifts towards 0.83-0.87 near the storm top for all cases.

The models do capture the statistical properties of the observed polarimetric variables to a certain extent, but the comparison also outlines many deficiencies in the synthetic polarimetric variables. The $Z_{DR}$ CFADs from the ensemble simulations exhibit
narrow distributions with peak values near zero above the melting layer, which does not differ among the three case studies. It also exhibits bimodal peaks below the melting layer compared to unimodal distribution in observations. Similar bi-modal CFADs were also reported by Matsui et al. (2019) for a simulated mesoscale convective system over Southern Great Plains, USA using both spectral bin microphysics and single moment cloud microphysics scheme, while the observed CFADs of $Z_{DR}$ exhibited a more smooth gradient below the melting layer as shown for the observation in this study as well. In their study, even
sensitivity studies with FO parameters also could not reproduce the distribution similar to the observations, while producing different results for the two microphysics schemes. In this study, the model tends to strongly underestimate the maximum reflectivities for case one but generally it exhibits a broader distribution of $Z_H$ for all three cases compared to the observations, with a peak around 30 dBZ above the melting layer. This higher reflectivity is caused by the dominance of graupel as discussed above. Consequently, the precipitation production by melting of graupel/hail below the melting level, as shown in the cross
sections of model simulated hydrometeors for all cases, could explain the second $Z_{DR}$ peak at approximately 2 dB in the lower levels. This possibly indicates that the modeled mechanism of precipitation formation below the melting layer differs from the observation. Furthermore, the use of a functional form of drop size distribution in the FO leading to a unique mapping between modeled quantities and synthetic polarimetric quantities can create errors (Kumjian et al., 2019), which could also be partly contributing to this bi-modal peak behaviour in the synthetic $Z_{DR}$ CFADs. Both the ensemble model runs and the observations
produce unimodal distribution for $K_{DP}$ peaking around 0.1 deg/km. However, the model again exhibits a narrower distribution above the melting layer compared to observation. Thus, the observed variability in $Z_{DR}$ and $K_{DP}$ above the melting layer is underestimated in the synthetic polarimetric variables. Part of this reduced variability can be explained by the deficiencies of the forward operator. Earlier, an extensive sensitivity study with the hydrometeor parameters in the same FO was conducted for a stratiform case over the same modelling domain (Shrestha et al., 2022). In their study, the model was found to exhibit a

low bias in the polarimetric moments above the melting layer, where snow was found to dominate, but none of the alternative shape and orientation setups for snow could provide sufficiently strong polarimetric signals to reproduce observed signals at these heights. The inability to reproduce the polarimetric characteristics of snow with T-Matrix also justifies the need for a scattering database. This issue needs to be revisited with more sophisticated forward operators available in the future (already planned in this project). For $\rho_{\mathrm{hv}}$, the CFADs are poorly simulated by the model, probably due to the shortcomings in forward operator assumptions on diversity of hydrometeor shapes and orientation (Shrestha et al., 2022). Although the synthetic $\rho_{\mathrm{hv}}$ exhibits very homogeneous high values above the melting layer, it does exhibit slightly reduced magnitude in locations with elevated $Z_{\mathrm{DR}}$. This pattern was found to consistent for all simulated case studies.

## 7  Conclusions

The TSMP model - in particular its atmospheric component COSMO with 2 moment cloud microphysics scheme - was found to generally underestimate the initial intensity of storms in terms of convective area fraction, extreme reflectivities. These underestimations were also reflected in the frequency distribution for high precipitation and also broader distribution of reflectivities. The model and FO were able to capture dominant polarimetric feature like $Z_{\mathrm{DR}}$ column but underestimated its width/magnitude compared to observations, and could not capture the collocated $K_{\mathrm{DP}}$ columns. Compared to observations, the model was able to simulate similar statistical distribution of $Z_{\mathrm{DR}}$ and $K_{\mathrm{DP}}$ but with less variability above the melting layer, while exhibiting bimodal distribution for $Z_{\mathrm{DR}}$ below the melting layer. The observations also additionally exhibited shifts in the peak of the $Z_{\mathrm{DR}}$ above the melting layer, which was not captured in the model simulations. This shift in the observations, could be associated with differences in partitioning of ice water content above the melting layer as well as the partial effect of attenuation correction algorithm.

The discrepancy between the observed and synthetic polarimetric feature could be attributed to the deficiency in the 2-moment cloud microphysics scheme, forward operator and to certain extent the attenuation correction algorithm or the radar data. Particularly, the model exhibits more graupel for all simulations, which also affects the precipitation production mechanism below the melting layer. While there is a strong understanding of polarimetric signatures for the raindrops, the mechanism by which the raindrops are produced and how the drop size distribution evolves, adds additional uncertainty.

For the 2-moment cloud microphysics scheme, the fixed CN concentrations and shape parameters of cloud drop size distribution could also be partly responsible for the overall too low storm intensities, thus regional measurements of CN/IN concentrations, surface precipitation and polarimetric radar data observations could be used together to constrain the shape parameters of cloud droplets. While regional measurements of CN/IN concentrations might not be readily available, sensitivity study with large scale aerosol perturbations or use of prognostic aerosol/trace gases module could be a way forward to minimize the uncertainty in polarimetric signatures due to aerosols.

On the forward operator for 2-moment cloud microphysics scheme, the water content of the ice hydrometeors can strongly modulate the dielectric constant and hence the scattering properties. This information is not directly available in the forward operator - and the melting parameterization in the FO does not completely compensate for the scattering properties of the ice

hydrometeors above the melting layer. So, future advancement in the FO should include parameterization for determining more accurate water content of the ice hydrometeors above the melting layer, which would help in obtaining more accurate dominant
polarimetric signatures.

Importantly, prominent polarimetric signature of convective storms like the $Z_{\mathrm{DR}}$ column appears to be poorly resolved at km-scale simulations. Future model evaluations with polarimetric radar data should focus on hyper-resolution simulations to better resolve the three-dimensional motion and microphysical processes associated with multivariate polarimetric signatures as well as uncertainty estimates in the attenuation correction of polarimetric moments for convective cases.

*Code and data availability.* The source codes for TSMP and the forward operator used in this study are freely available from https://www.terrsysmp.org/ and https://git2.meteo.uni-bonn.de/git/pfo respectively with registration. The codes for radar calibration and attenuation correction is available from https://github.com/meteo-ubonn/miubrt. The data used for the model runs including initial conditions for the soil-vegetation states are available from Deutscher Wetterdiest (https://www.dwd.de/DE/leistungen/pamore/pamore.html) and https://doi.org/10.5880/TR32DB.40 respectively

.

**Abbreviations**

**Aerosol Specification**

$\epsilon_s$      Solubility of aerosol

$\log(\sigma_s)$ Logarithm of the geometric standard deviation of aerosol

$N_{cn}$      Condensation nuclei (CN) concentration $[m^{-3}]$

$N_{x=c,r,i,s,g,h}$ Concentration of hydrometeors: cloud(c), rain(r), ice(i), snow(s),graupel(g) and hail(h) $[m^{-3}]$

$N_{x=d,s,o}$ Ice nuclei concentration for dust (d), soot (s) and organics (o) $[m^{-3}]$

$q_{x=c,r,i,s,g,h}$ Mixing ratio of hydrometeors:cloud(c), rain(r), ice(i), snow(s),graupel(g) and hail(h) $[kg/kg]$

$R_2$      Mean radius of the dominant mode of the aerosol size distribution $[\mu m]$

**Models**

B-PRO   Bonn Polarimetric Radar Forward Operator

CLM     NCAR Community Land Model

COSMO  Consortium of Small-scale Modelling

COSMO-DE  High resolution ( 2.8 km) configuration of the COSMO model over Germany(DE)

COSMO-DE EPS  COSMO-DE Ensemble Prediction System

EMVORADO  Efficient Modular Volume Scan Radar Operator

GFS      Global Forecast System of NCEP

GME      Global Model of DWD

IFS      Integrated Forecast System of ECMWF

OASIS3-MCT  Ocean Atmosphere Sea Ice Soil, version 3.0 - Model Coupling Toolkit

ParFlow  Parallel Flow hydrologic model

TSMP   Terrestrial Systems Modelling Platform (COSMO, CLM and ParFlow coupled using OASIS3-MCT)

UM      Unified Model of the UK Met Office

**Polarimetric variables**

$\delta$        Backscatter differential phase

$\Phi_{\mathrm{DP}}$     Total differential phase shift

$\rho_{\mathrm{hv}}$     Cross-correlation coefficient between horizontally and vertically polarized return signals

$\sigma_c$       Width of canting angle distribution (The canting angle is the angle between the horizontal and the symmetry axis of

the falling particles (horizontally aligned particles have a $0°$ canting angle). In a radar observed volume containing

several particles, canting angles vary from particle to particle giving rise to a distribution. The width of the canting

angle distribution is a measure of the variability of canting angles in that sample.)

$\varphi_{\mathrm{DP}}$     Propagation differential phase shift

$AR$     Aspect ratio (Ratio between the horizontal and the vertical dimension of the particle)

$D_x$      Equivalent/Maximum diameter of spherical/non-spherical particle

$K_{\mathrm{DP}}$     Specific differential phase $[degkm^{-1}]$

$Z_{\mathrm{DR}}$     Differential reflectivity $[dB]$ (It is the ratio of reflectivity for horizontal and vertical polarization in linear units)

$Z_{\mathrm{H}}$      Reflectivity for horizontal polarization $[dBZ]$

## Appendix A

**Table A1.** Estimated biases for $Z_H$ and $Z_{DR}$ for both radars and for each event

|  | BoXPol $Z_H$ [dBZ] | JuXPol $Z_H$ [dBZ] | BoXPol $Z_{DR}$ [dB] | JuXPol $Z_{DR}$ [dB] |
| --- | --- | --- | --- | --- |
| **5 July 2015** | -3 | -7 | -1.4 | -2.3 |
| **13 May 2016** | -0.9 | -5 | -1 | -1.95 |
| **6 July 2017** | -0.5 | -7 | -0.8 | -2.5 |

*Author contributions.* PS designed the study, conducted the model simulations and forward operator calculations, carried out the analysis, wrote the paper and obtained the grant for the study. ST aided in initial conceptualization of the study. RE processed the radar data (calibration and attenuation correction) for model comparison. ST, RE, CS aided with the analysis of the radar data.

*Competing interests.* The authors declare that they have no conflict of interest.

*Acknowledgements.* The research was carried out in the framework of the priority programme SPP-2115 "Polarimetric Radar Observa-
tions meet Atmospheric Modelling (PROM)" in the project ILACPR funded by the German Research Foundation (DFG, Grant SH 1326/1-1). We gratefully acknowledge the computing time (project HBN33) granted by the John von Neumann Institute for Computing (NIC) and provided on the supercomputer JUWELS at Jülich Supercomputing Centre (JSC). The pre-processing and post-processing of input data was done using the NCAR Command language (Version 6.4.0). The analysis of the radar data was done using the wradlib libraries (https://docs.wradlib.org/en/stable/index.html)

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

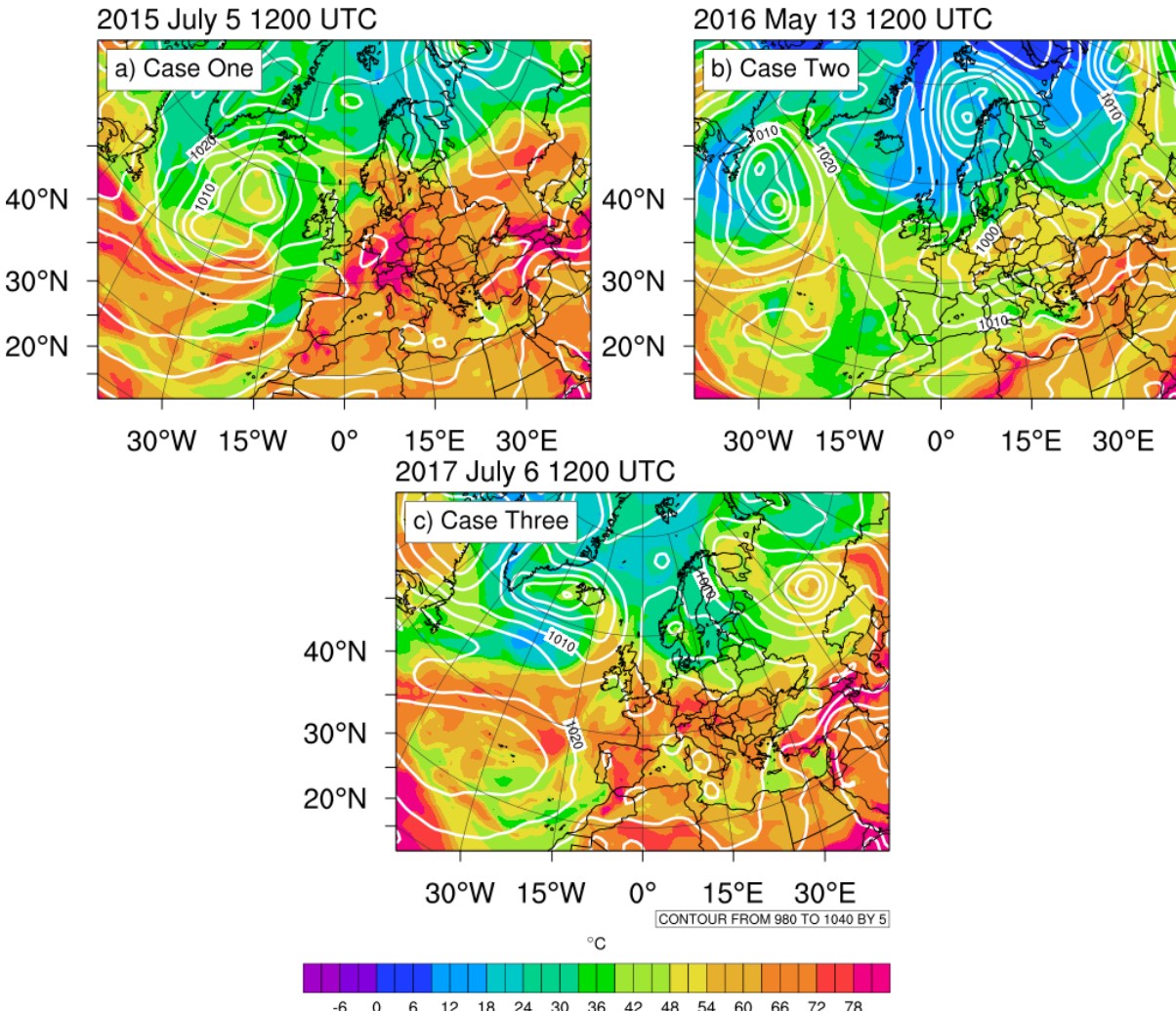

**Figure 1.** Synoptic conditions for the three different cases - surface pressure reduced to mean sea level and 850 hPa pseudo-equivalent potential temperature. The plots are based on GFS analysis data.

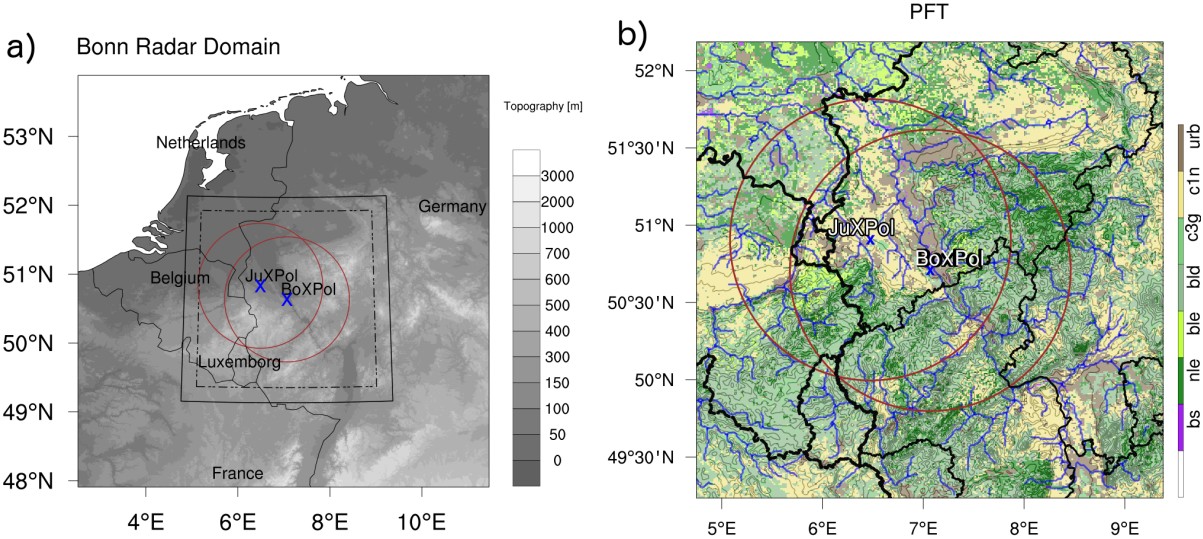

**Figure 2.** a) Spatial pattern of topography and extent of Bonn Radar domain (solid line) including the coverage of BoXPol and JuXPol (red circles). The dotted lines indicate the inner domain (excluding the relaxation zone) used to compute the domain average precipitation. b) Spatial pattern of plant functional types (PFTs). Also shown is the coverage of two X-band radars.

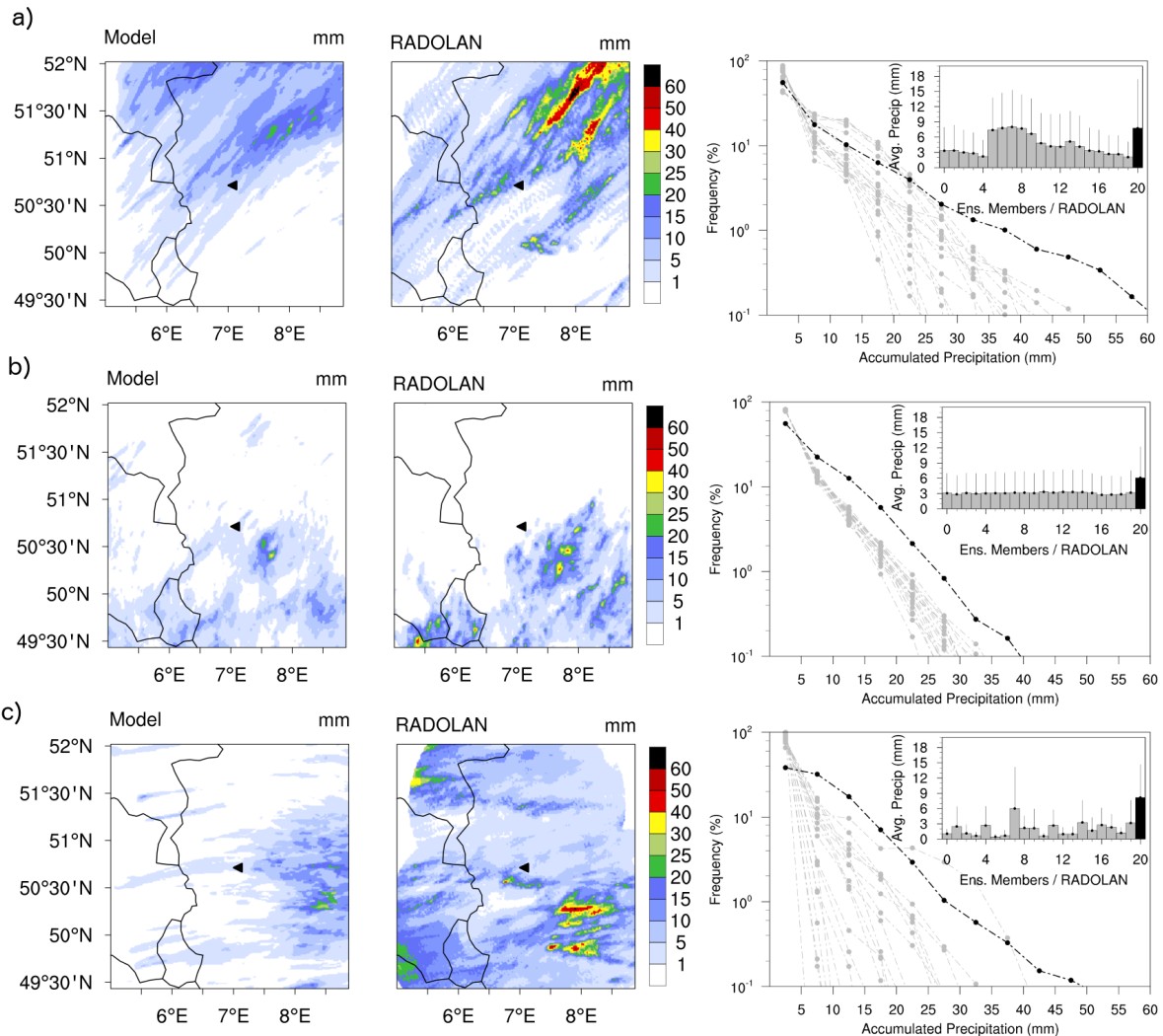

**Figure 3.** Spatial pattern and frequency distributions of accumulated precipitation over the Bonn Radar domain for three case studies (a,b and c). For each case studies, the left and middle panel shows the spatial pattern of accumulated precipitation from model (ensemble average) and observations. The right panel shows the frequency distributions of accumulated precipitation for each ensemble member (light grey dashed line) and observation (black dashed line). The inset in the right panel shows the domain average accumulated precipitation for each ensemble member (light grey color bar) and observation (black color bar) with one standard deviation (solid line above the bars).

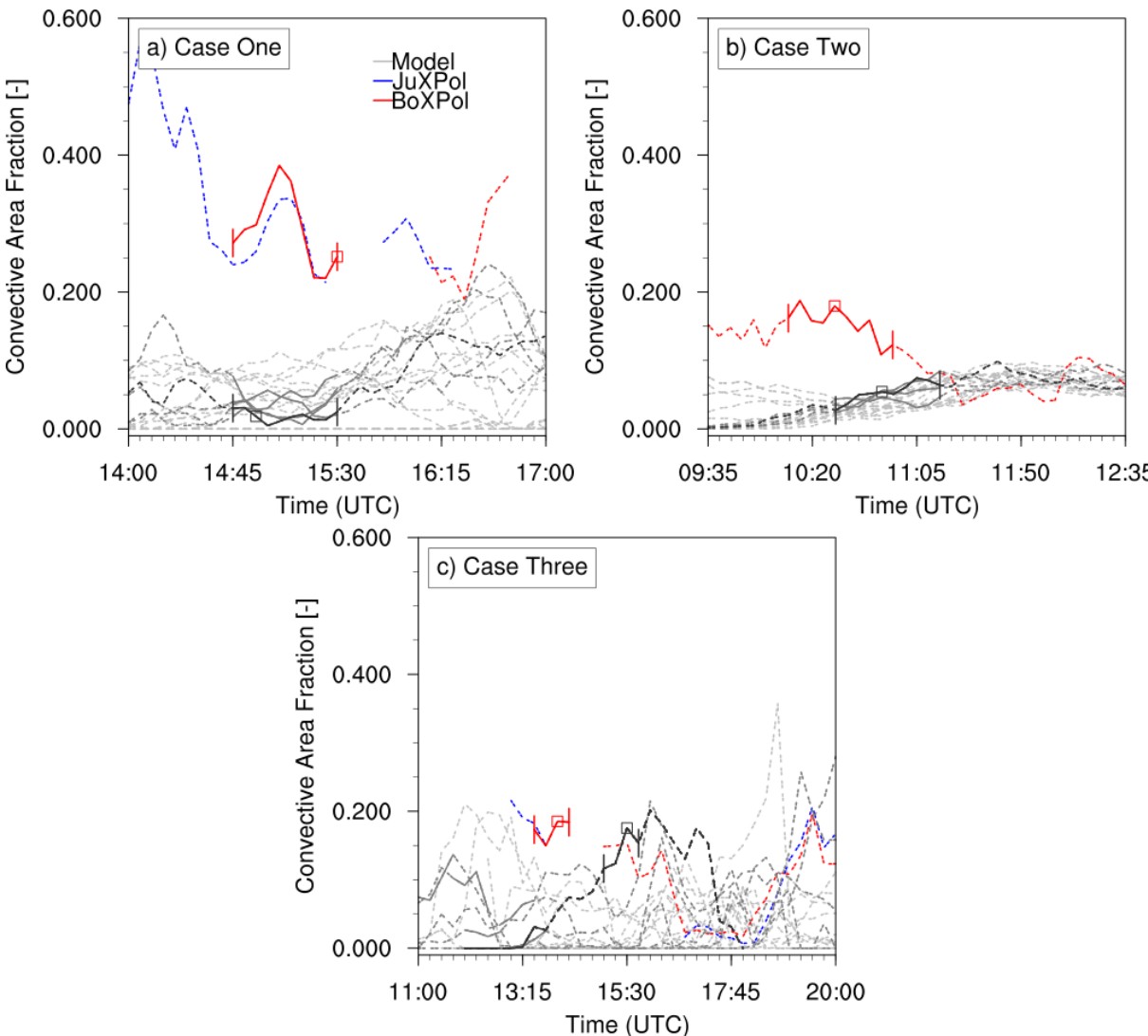

**Figure 4.** Convective Area Fraction (CAF) of model ensemble members and observations for the three different case studies. The two vertical bars defines the time-period used to compute CFADs for observation (red color) and model (gray color) with selected ensemble members (soild lines within this extent). The ensemble member with solid black line is used for polarimetric signature comparison. The square marker (red and gray) represents the snapshot used for polarimetric comparision between observation and model for each case study. The observations from BoXPol or JuXPol are shown upon coverage and data availability. The gaps in the radar data represents times, when the polarimetric signatures are strongly attenuated or if the storm extent is only partially covered by the radar.

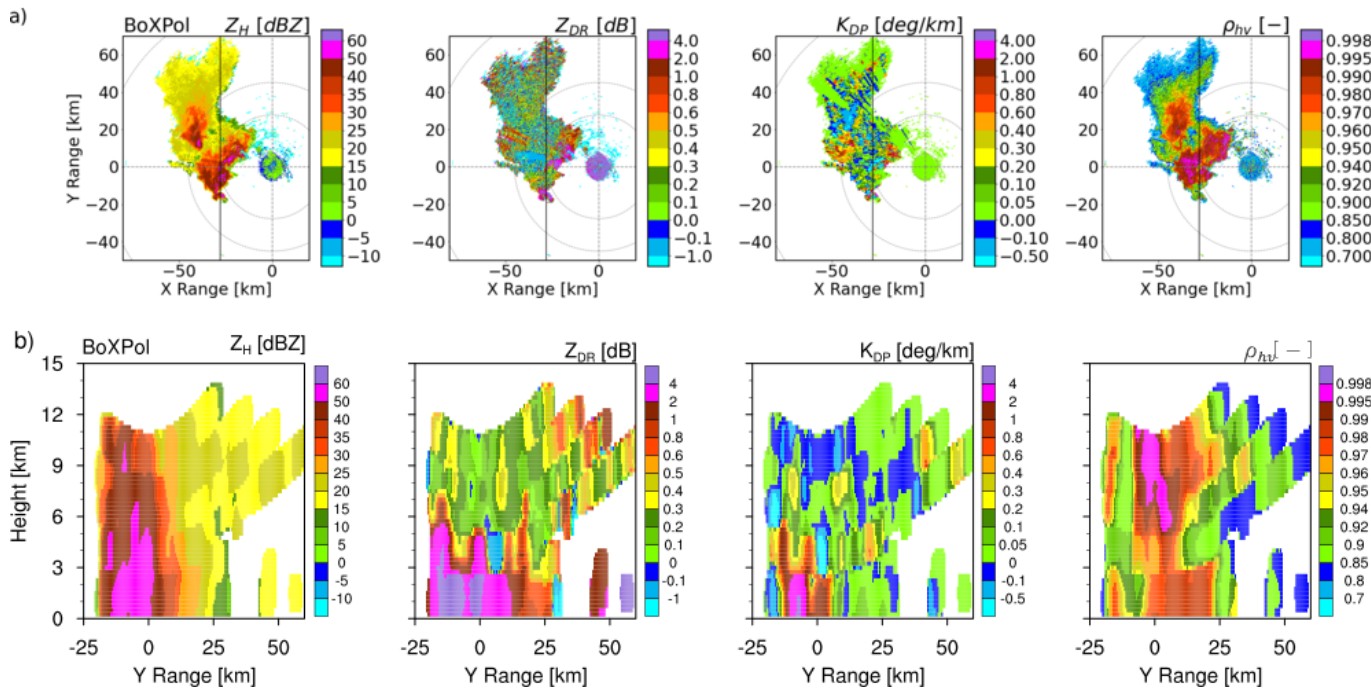

**Figure 5.** a) Plan position indicator (PPI) plots of horizontal reflectivity, differential reflectivity, sp. differential phase and cross-correlation coefficient at 8.2 degree elevation measured by BoXPol on 5 July 2015 at 1530 UTC. The dotted gray circles represent slant ranges for the chosen elevation angle, associated with heights of 1 km (lower levels) , 4.5 km (melting layer) and 7 km (upper levels). b) Cross-section of the same polarimetric variables from the gridded data. The vertical solid black line along the Y Range in a) indicates the location of cross-section plots.

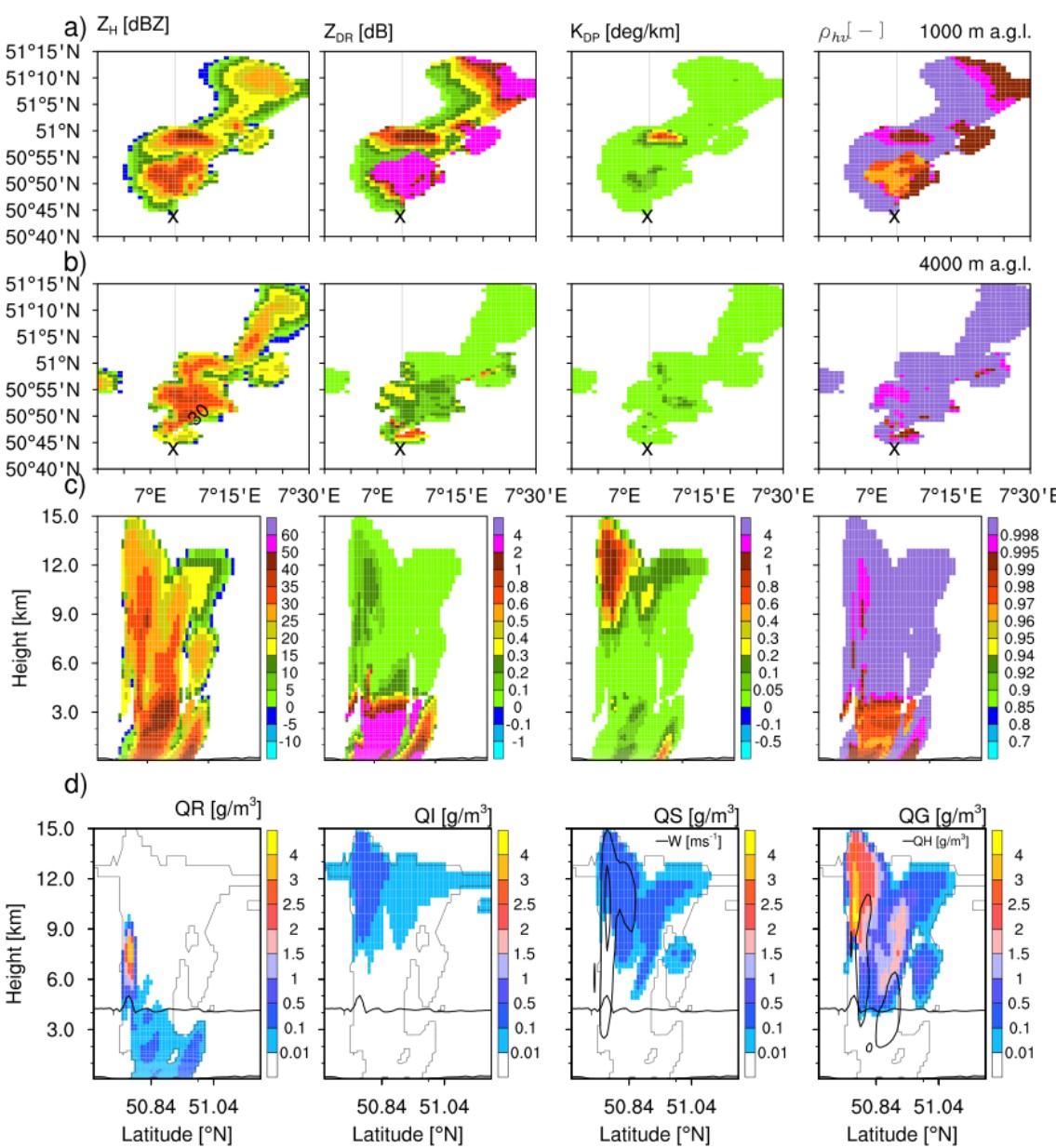

**Figure 6.** a,b) Model simulated horizontal reflectivity, differential reflectivity, sp. differential phase and cross-correlation coefficient at low level (1000 m a.g.l.) and near melting layer (4000 m a.g.l.) on 5 July 2015 at 1455 UTC. The 'x' mark refers to the BoXPol location. The gray solid line indicates the location of cross-section. c) Cross-section of the same polarimetric variables. d) Cross-section of model simulated hydrometeor density [QR(rain), QI (ice), QS (snow), QG (graupel) and QH (hail)]. Also shown are the $0°C$ line (solid black line) indicating the melting layer, contours of vertical velocity [5, 40 m/s] with QS and contours of hail mixing ratio with QG.

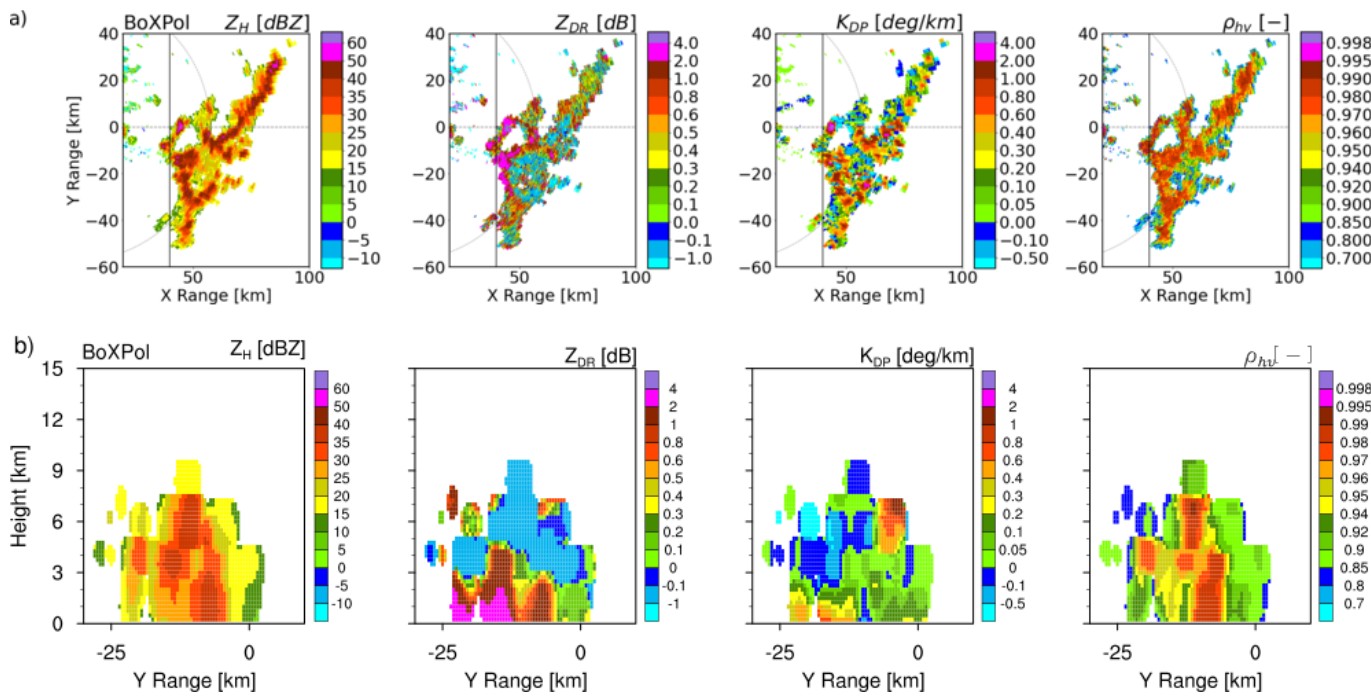

**Figure 7.** a) Plan position indicator (PPI) plots of horizontal reflectivity, differential reflectivity, sp. differential phase and cross-correlation coefficient at 1.0 degree elevation measured by BoXPol on 13 May 2016 at 1030 UTC. The dotted gray circles represent slant ranges for the chosen elevation angle, associated with height of 1 km (lower levels). b) Cross-section of the same polarimetric variables from the gridded data. The vertical solid black line along the Y Range in a) indicates the location of cross-section plots.

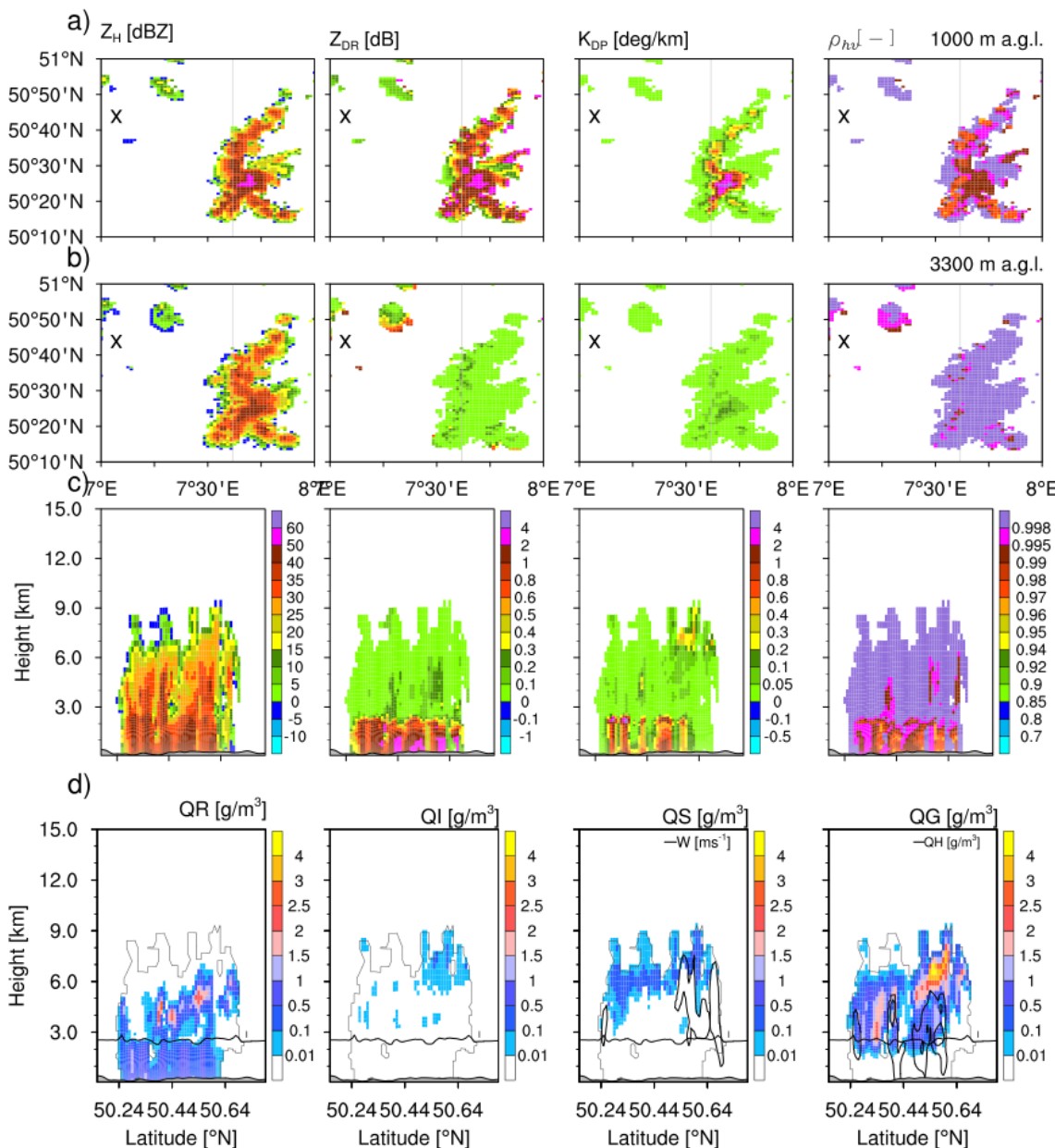

**Figure 8.** a,b) Model simulated horizontal reflectivity, differential reflectivity, sp. differential phase and cross-correlation coefficient at low level (1000 m a.g.l.) and near melting layer (3300 m a.g.l.) on 13 May 2016 at 1050 UTC. The 'x' mark refers to the BoXPol location. The gray solid line indicates the location of cross-section. c) Cross-section of the same polarimetric variables. d) Cross-section of model simulated hydrometeor density [QR(rain), QI (ice), QS (snow), QG (graupel) and QH (hail)]. Also shown are the $0°C$ line (solid black line) indicating the melting layer, contours of vertical velocity [5, 40 m/s] with QS and contours of hail mixing ratio with QG.

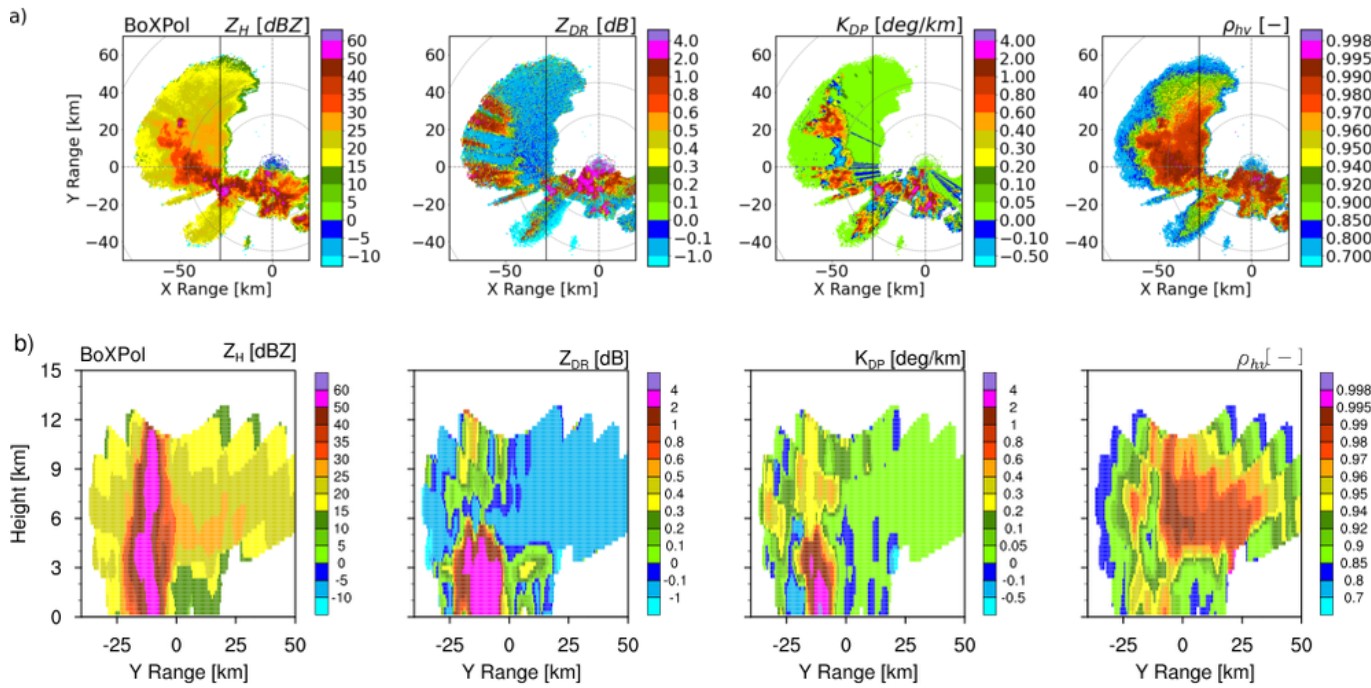

**Figure 9.** a) Plan position indicator (PPI) plots of horizontal reflectivity, differential reflectivity, sp. differential phase and cross-correlation coefficient at 8.2 degree elevation measured by BoXPol on 6 July 2017 at 1400 UTC. The dotted gray circles represent slant ranges for the chosen elevation angle, associated with height of 1 km (lower levels), 4 km (melting layer), 6.5 km (upper levels) and 13 km. b) Cross-section of the same polarimetric variables from the gridded data. The vertical solid black line along the Y Range in a) indicates the location of cross-section plots.

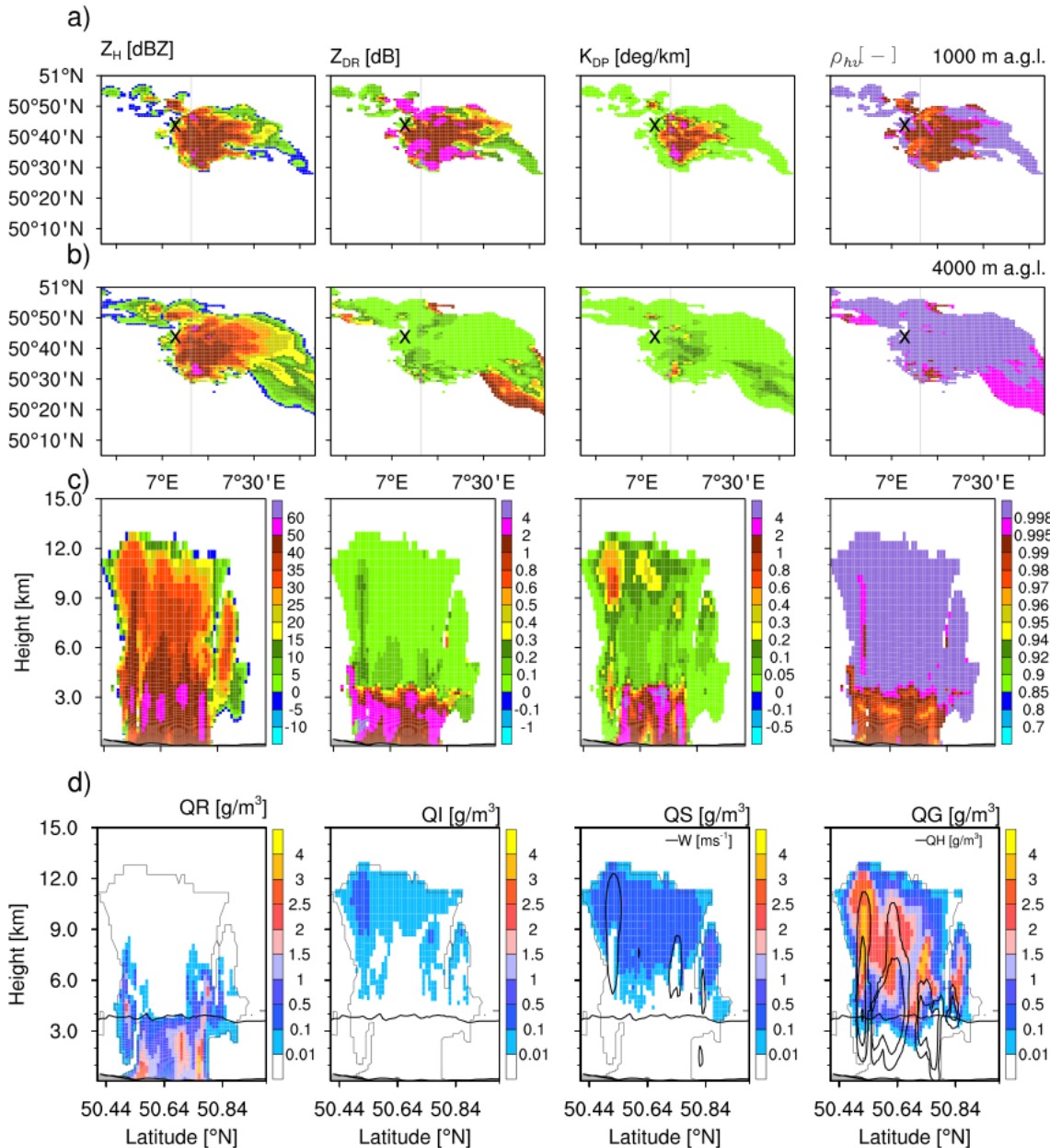

**Figure 10.** a,b) Model simulated horizontal reflectivity, differential reflectivity, sp. differential phase and cross-correlation coefficient at low level (1000 m a.g.l.) and near melting layer (4000 m a.g.l.) on 6 July 2017 at 1530 UTC. The 'x' mark refers to the BoXPol location. The gray solid line indicates the location of cross-section. c) Cross-section of the same polarimetric variables. d) Cross-section of model simulated hydrometeor density [QR(rain), QI (ice), QS (snow), QG (graupel) and QH (hail)]. Also shown are the $0°C$ line (solid black line) indicating the melting layer, contours of vertical velocity [5, 40 m/s] with QS and contours of hail mixing ratio with QG.

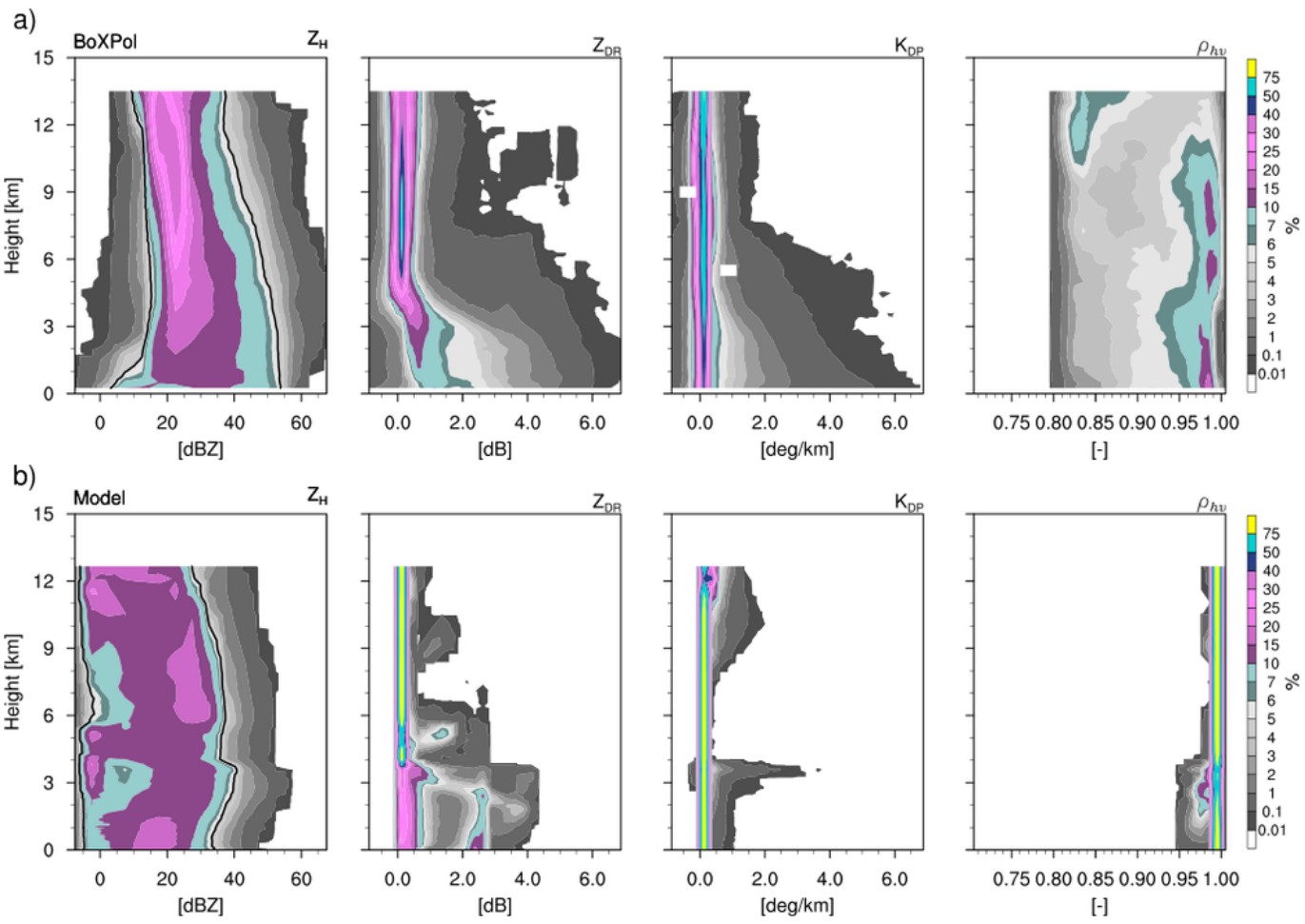

**Figure 11.** Contoured frequency altitude diagrams (CFADs) of horizontal reflectivity, differential reflectivity, sp. differential phase and cross-correlation coefficient from 1445 to 1530 UTC on 5 July 2015. CFADs from the model are shown for 5 ensemble members.

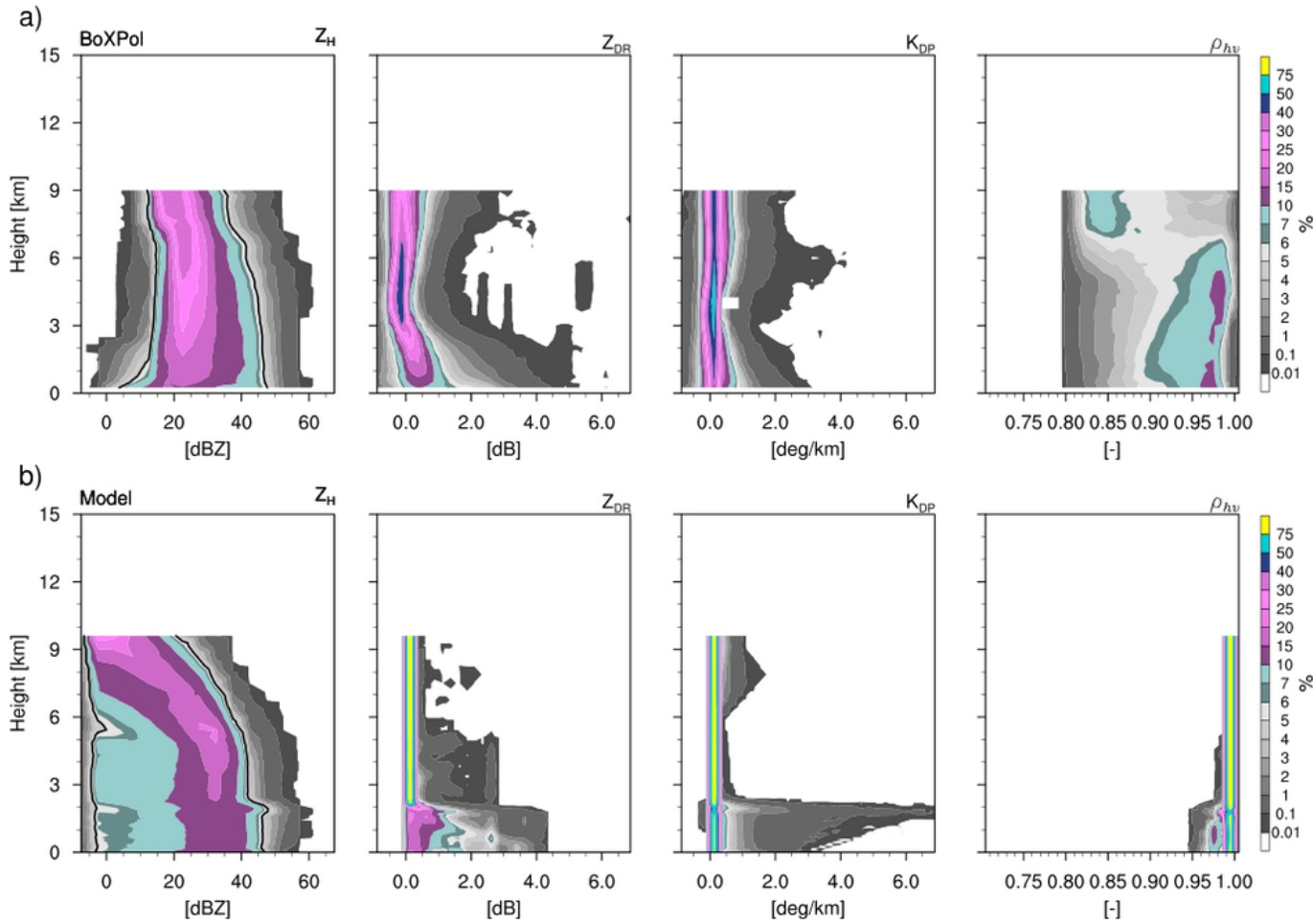

**Figure 12.** Contoured frequency altitude diagrams (CFADs) of horizontal reflectivity, differential reflectivity, sp. differential phase and cross-correlation coefficient from 1010 to 1055 UTC on 13 May 2016. CFADs from the model are shown for 5 ensemble members from 10:30-11:15 UTC.

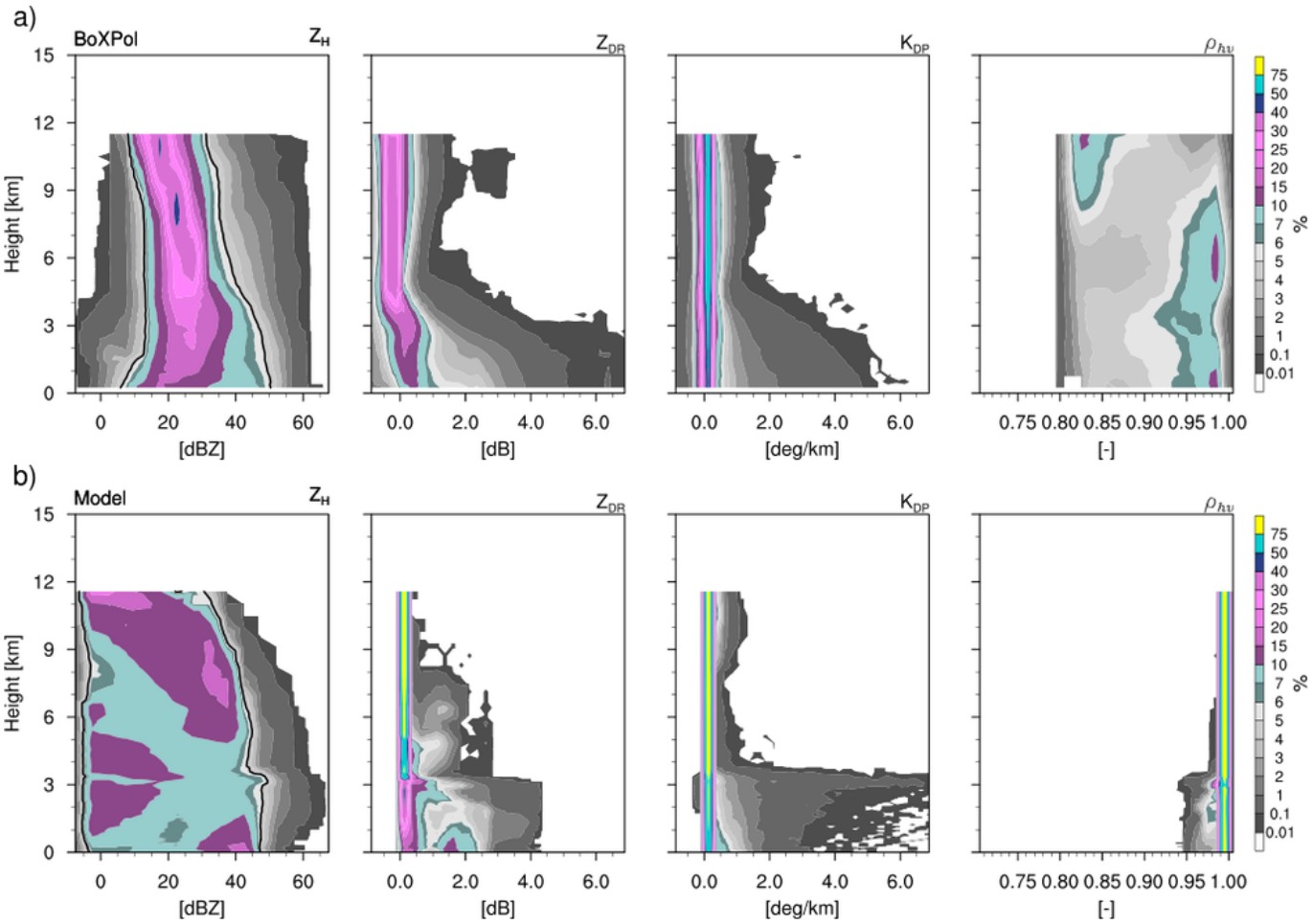

**Figure 13.** Contoured frequency altitude diagrams (CFADs) of horizontal reflectivity, differential reflectivity, sp. differential phase and cross-correlation coefficient from 1330 to 1415 UTC on 13 May 2016. CFADs from the model are shown for 1 ensemble member from 1500 to 1545 UTC.

TABLES

**Table 1.** Hydrometeor parameters for mass-diameter relationship and generalized gamma distribution for the of 2-moment microphysics scheme including minimum and maximum values of mean particle mass.

| Hydrometeors | a $(mkg^{-b})$ | b | $\nu$ | $\mu$ | $x_{min}$ (kg) | $x_{max}$ (kg) |
|---|---|---|---|---|---|---|
| cloud | 0.124 | 1/3 | 0.0 | 1/3 | $4.20 \times 10^{-15}$ | $2.60 \times 10^{-10}$ |
| rain | 0.124 | 1/3 | 0.0 | 1/3 | $2.60 \times 10^{-10}$ | $3.00 \times 10^{-6}$ |
| ice | 0.835 | 0.39 | 0.0 | 1/3 | $1.00 \times 10^{-12}$ | $1.00 \times 10^{-6}$ |
| snow | 2.4 | 0.455 | 0.0 | 0.50 | $1.00 \times 10^{-10}$ | $2.00 \times 10^{-5}$ |
| graupel | 0.142 | 0.314 | 1.0 | 1/3 | $1.00 \times 10^{-9}$ | $5.00 \times 10^{-4}$ |
| hail | 0.1366 | 1/3 | 1.0 | 1/3 | $2.60 \times 10^{-9}$ | $5.00 \times 10^{-4}$ |

**Table 2.** Large-scale continental aerosol specification for cloud droplet nucleation and default parameters for ice nucleation.

| | $N_{CN}, m^{-3}$ | $R2, \mu m$ | $log(\sigma_s)$ | $\epsilon_s$ | $N_{x=d}, m^{-3}$ | $N_{x=s}, m^{-3}$ | $N_{x=o}, m^{-3}$ |
|---|---|---|---|---|---|---|---|
| $CD^1$ | $1700 \times 10^6$ | 0.03 | 0.2 | 0.7 | $162 \times 10^3$ | $15 \times 10^6$ | $177 \times 10^6$ |

**Table 3.** Assumed hydrometeor physical properties for T-matrix computation in the B-PRO

| | $D_x$ | AR | $\sigma_c$ |
|---|---|---|---|
| Rain | 50 $\mu$m-8 mm | (Brandes et al., 2002) | 10° |
| Cloud ice | 20 $\mu$m- 0.5 mm | $\sim 0.2, plates$ (Andrić et al., 2013) | 12° |
| Snow | 50 $\mu$m $-$ 20 mm | $0.7 - 0.2 \times D_x/D_{x,max}$ (Xie et al., 2016) | 40° |
| Graupel | 50 $\mu$m $-$ 30 mm | $max(1.0 - 20 \times D_x, 0.8)$ (Ryzhkov et al., 2011) | 40° |
| Hail | 50 $\mu$m $-$ 30 mm | $max(1.0 - 20 \times D_x, 0.8)$ (Ryzhkov et al., 2011) | 40° |