# Peer review of "Evaluation of modeled summertime convective storms using polarimetric radar observations"

_Atmospheric Chemistry and Physics, 2021_

## Author Response (AR1)

**Reply to RC1**

General comments:

   The paper presents an interesting study that statistically compares polarimetric radar observations from three storm systems in northwestern Germany to ensemble modeling results. I enjoyed reading the paper and found the results to be interesting, valuable, and very worthy of publication. I do, however, have several comments that I feel would improve the manuscript. I will address these in more detail in the next section.

**We are very thankful for the reviewer's comments. Below, we address the reviewer's specific comments (in bold blue).**

Individual comments:

1) While I do have a background in polarimetric radar observations and, to some degree, the use of polarimetric radar data in numerical models, I am unfamiliar with most of the models used in this study. Upon my first reading, I must admit that I was a little overwhelmed by the numerous acronyms that were being introduced and found myself continually going back to refresh my memory. After gett8ng a few pages in, I skipped to the back of the paper to see if there was perhaps an appendix that summarized the list of acronyms. Is this something the authors might consider?

**We think it's a valid suggestion. In the revised manuscript, we have moved much of the acronyms to the Abbreviation section, which has improved the text flow in the manuscript. The acronyms are also better described here.**

2) In section 3.2, I really don't feel like I had a good understanding of how the 20 ensemble members for each case were obtained. That is, it states the 20 ensemble members represent "uncertainties in model physics and lateral boundary conditions by combining five model physics perturbations with four global models are used for the initial and lateral boundary conditions". I found this description to be a little vague. Can you be more specific about what those five model physics perturbations and, more importantly, four global models are? And the results from those runs are used as the initial and boundary conditions for the COSMO runs? Also, what does COSMO-DE stand for? COSMO is introduced earlier but, looking back into the paper, I was unable to find what COSMO-DE referred to.

**COSMO-DE is a high resolution (~2.8 km) configuration of the COSMO model encompassing the entire extent of Germany including some neighboring countries; thus "DE" is the code for Germany (Deutschland). The 20 ensemble members of COSMO-DE EPS can be divided into 4 subsets of 5 members each. The 4 subsets represent the different global models used for lateral boundary conditions and initialization: the Integrated Forecast System of ECMWF (IFS) [Janssen and Bidlot, 2002], the global model of DWD (GME) [Majewski et al., 2002], the Global Forecast System of NCEP (GFS) [Environmental Modeling Center, 2003] and the Unified Model of the UK Met Office (UM) [Staniforth et al., 2006]. Each of the 5 members in the subsets use different parameter sets in their parameterizations. These parameter sets govern the entrainment rate for shallow convection, the critical value for normalized oversaturation, the scaling factor of the laminar boundary layer for heat, and the asymptotic mixing length of turbulence. For details we refer to Gebhardt et al. (2011) and Peralta et al. (2012), which are also cited in the manuscript.**

**The hourly output from the COSMO-DE EPS provided by DWD is then used for the model runs in this study. For clarity, the description in Section 3.2 has been modified:**

**Ln 154: "The hourly model output from the 20 ensemble members of the COSMO-DE Ensemble Prediction System (EPS: Gebhardt et al. 2011; Peralta et al. 2012) provided by DWD is used for the model runs in this study. The COSMO-DE**

**is a high resolution (~2.8 km) configuration of the COSMO model encompassing the entire extent of Germany. The 20 ensemble members of COSMO-DE EPS can be divided into 4 subsets of 5 members each. The 4 subsets represent different global models: the Integrated Forecast System of ECMWF (IFS) [Janssen and Bidlot, 2002], the global model of DWD (GME) [Majewski et al., 2002], the Global Forecast System of NCEP (GFS) [Environmental Modeling Center, 2003] and the Unified Model of the UK Met Office (UM) [Staniforth et al., 2006], used to vary the boundary conditions of the COSMO-DE. Each subset of the 5 members is then perturbed by varying a set of parameters that control the physics parameterization of the COSMO model. "**

At the beginning of section 5.3, there is a short discussion of clustering. Clustering, I believe, refers to how combined plots of two polarimetric variables will cluster in multidimensional space. This seems totally unrelated to computing convective area fractions of a single radar variable, such as reflectivity. Also, can the authors provide a more complete description of convective area fraction (CAF)? I know this is a concept that has been used in numerous papers, but without description I am often left wondering if the convective area fraction is with respect to the grid being used, or with respect to all reflectivity points (for example) above a certain dBZ threshold? It seems to me that a CAF can be defined in many different ways. Also, how is CAF impacted if, for example, a portion of the system that is being studied is moving off of the grid over which the CAF is being sampled?

**Yes, the reviewer is correct, here clustering refers to how multiple polarimetric variables will cluster in multidimensional space. This clustering also depends on the stages of storm evolution. So, here we use the temporal evolution of the convective area fraction to identify the development stage of the storm in the measurements and in the synthetic radar data. This is then used to minimize effects of the mismatches in space and time when comparing both.**

**In this study the convective area fraction (CAF) is the area fraction of the storm with $Z_H > 40$ dBZ at 2 km height a.g.l. divided by the total area of the storm which encompasses the grid points of the storm with $Z_H > 0$ dBZ at the same height. Only CAF evolutions were compared for which the storm stayed within the domain. This restriction affected e.g., Case 1 and 2 when for some members the storm approached the boundaries in the last 30 minutes. Due to extended sampling time used in Case 3 the compared time interval is reduced because the storm moved off the grid in the simulations.**

**The following text has been added in the revised manuscript:**

**Ln 270: "The total area of the storm for CAF estimate, includes the grid points of the storm with radar reflectivity >0 dBZ at 2 km height a.g.l. The time extent of the CAF evolution was chosen such that the storm is within the domain. However, due to variability in the ensemble members, some members are affected as part of the storm approaches the boundary in the last 30 minutes of CAF evolution for Case 1 and 2. And, due to extended sampling time used in Case 3, the CAF is partly impacted by the storm moving off the grid for the synthetic data."**

3) In sections 5.3.1, 5.3.2, and 5.3.3, I am confused why the elevation angle 8.2 is used for a PPI for cases 1 and 3 (Figs. 5 and 9) and an elevation angle of 1.0 is used for case 2 (Fig. 7). Using an elevation angle of 8.2 for a PPI seems very unusual. Please explain why such high elevation angles are being used for these plots.

**An 8.2° elevation angle for a PPI is unusual for near-surface quantitative precipitation estimation for which we use indeed PPIs measured at 0.5° or 1° elevation angle or terrain-following scans. In recent years, however, volume scans consisting of a series of PPIs measured at different elevations, mostly between 0.5° and 30°, became more popular in order to get a 3D picture of hydrometeors and microphysical processes e.g. for improved process understanding, model evaluation and data assimilation. Such volume scans also enable us to construct vertical cross-sections of convective systems (e.g. Fig. 5b). Choosing a PPI measured at higher elevations for Fig. 5a and Fig. 9a gives**

insights also of the measurements at different heights (~1 km, near melting layer and 2~3 km above melting layer) of the deep convective systems (radar measures at increasing height with increasing distance from the radar). Together with the spatial extent and location of the system, these figures complement the cross-section in Fig. 5b and Fig. 9b. Case 2 instead, was less vertically extensive and further distant from the radar, so 1° scan was used to explore the low-level features.

For clarity on the use of the different elevation scans, we have added the following text in Section 3.3 of the revised manuscript:

Ln 182: "Both X-band Doppler radars produce volume scans consisting of a series of Plan Position Indicator scans (PPIs) measured at different elevations, mostly between 0.5° and 30°. The use of these multiple sweeps became more popular in recent years in order to get a 3D picture of the spatial distribution of hydrometeors and microphysical processes. These PPIs can be exploited for improved process understanding, model evaluation and data assimilation. And, such volume scans also enable us to construct vertical cross-sections of the convective systems."

Ln 211: "Based on the time and the distance of the storm from the radar for the different cases, PPIs measured at different elevation are used - to provide optimal insights in convective systems at different heights (~1 km, near melting layer and 2~3 km above melting layer)"

4) Overall comment on the figures, my philosophy has always been that figure captions should contain enough information that they could be "stand alone", i.e., that the reader should be able to fully interpret the figure without having to refer back to the text. That being said, I feel that much can be done in this manuscript to improve figure captions and, in a few cases, the figures as well. As an example, I felt that the caption describing the right most panel of Fig. 2 could have been much better, particularly the description of the rightmost panel.

The caption for Fig. 2 has been improved in the revised manuscript:

…"The right panel shows the frequency distribution of accumulated precipitation for each ensemble member (light grey dashed line) and observation (black dashed line). The inset in the right panel shows the domain average accumulated precipitation for each ensemble member (light grey color bar) and observation (black color bar) with one standard deviation (solid line above the bars)."

The captions for the remaining figures has also been improved, where applicable.

5) The text states that there were 104 GRDC stations, 36 were considered useful for this study. These figures show 22 or 23 stations, but the figure is very crowded with all of the stations grouped together in the rightmost 2/3rds of the figure with lots of unneeded and wasted white space on the left, etc. If several of the stations are not going to be used (presumably those that were to be plotted on the left side of each of the figures), I would suggest eliminating the "which space" and making the figure easier to read.  Also, are there 20 asterisks representing the 20 ensembles plotted for each station with the some of them just overlayed on each other?

The model evaluation with GRDC station data has been removed from the revised manuscript to keep the focus on the model evaluation with polarimetric radar data. This was suggested by reviewer #2.

6) Figure 4: Time labels need to be improved.

**We assume that the reviewer here refers to the missing second digit in the minutes of some labels. This has been fixed in the revised manuscript.**

Specific comments:

1) Overall, the paper is well written. There are some very minor grammatical issues throughout the text. I'll make just a few suggestions here.

**We are again very thankful for the reviewer's comments. Below, we address the reviewer's suggestions.**

2) There is also some inconsistency throughout the paper on whether the word "modeled"and "modeling" should be spelled with one "l" or two "ll''s.

**Fixed. We have now used "modelling" consistently.**

3) Line 7: Remove "however".

**Corrected.**

4) Line 9: Suggest replacing "besides" with "in addition to".

**Corrected.**

5) Line 20: Suggest rewording from "Polarimetric radar observations provide ZDR,…" to"Besides ZDR, polarimetric radar observations provide…"

**Corrected.**

6) Line 32: Suggest changing "thus e.g. provides insight on new snow generation" to"thereby providing insight into the generation of new snow".

**Corrected.**

7) Line 33: Suggest changing "measure for the diversity" to "measure of the diversity".

**Corrected.**

8) Line 34: Change "These informations" to "This information".

**Fixed.**

9) Line 47: Remove "e.g."

**Fixed.**

10) Line 49: Suggest changing "operator and due" to "operator due".

**Here, the "uncertainty in the model evaluation in radar space" stems from the "uncertainty in the assumptions made in the forward operator" and the "uncertainties of polarimetric radar measurements".**

11) Line 50: Remove "e.g."

**Fixed**

**Reply to RC2**

**General comments:**

This study utilizes ensemble Terrestrial Systems Modeling Platform (TSMP) simulations with forward dual-pol radar operator to evaluate the performance of simulated cloud microphysical processes for three summertime convective storms over northern Germany using bias corrected X-band radar observations. The paper presents some interesting results and contributes to scientific community in this field. However, the methodologies (section 2 and 3) need to be more clearly written and reorganized to be published. In addition, there are many grammatical errors and typos that require corrections. Thus, the reviewer suggests the manuscript to be reconsidered after major revisions are made with the following conditions.

**We are very thankful for the reviewer's comments. Below, we address the reviewer's specific comments (in bold blue), which has helped to improve Section 2 and 3 in the revised manuscript.**

**Major comments (scientific questions/issues):**

1) The abstract is too general and does not provide a solid conclusion based on the study results. Please revise the abstract.

**The abstract has now been revised. We have added the following sentence to better reflect the findings from the study:**

**"Features like column of enhanced differential reflectivity ($Z_{DR}$ column, which is a proxy for updraft), size sorting and aggregation that are observed and/or inferred from the radar data are captured by the model. Above the melting layer, the model exhibits low variability in polarimetric variables compared to observations. Below the melting level, the model does capture the increase in reflectivity, $Z_{DR}$ and specific differential phase ($K_{DP}$) as in the observations. "**

**"The contoured frequency altitude diagrams (CFADs) of $Z_{DR}$ and $K_{DP}$ were similar but the model exhibits a relatively narrow distribution above the melting layer for both, and a bi-model distribution for $Z_{DR}$ below the melting layer. The CFAD of the cross-correlation coefficient ($\rho_{hv}$) was poorly simulated."**

2) The authors introduce many notations in sections 1 and 2 without proper explanation oractual use of the equations. e.g. backscatter differential phase (delta), aerosol size distribution (R2), the logarithm of its geometric standard deviation log(sigma), solubility (epsilon), aspect ratio (AR), width of canting angle distributions (sigma), etc.

Please provide proper equations to these notations.

**In the revised manuscript, we have moved much of the notations from Section 1 and 2 to the Abbreviations, which has improved the text flow in the manuscript. The acronyms are also better described in this section. Further, the text has been revised in the manuscript to better describe the parameters, which do not need additional equations.**

**The backscatter differential phase is better described in the revised manuscript:**

**Ln 57: "In contrast, $K_{DP}$ is not affected by miscalibration and attenuation. However, the total differential phase shift is a combination of backscatter differential phase ($\delta$) and propagation differential phase ($\phi_{DP}$); thus, the subtraction of the former from the total differential phase shift ($\Phi_{DP}$) is required before computing $K_{DP}$. This is particularly important when hydrometeor sizes are in the range of or larger than the radar wavelength; these so-called resonance**

**effects are most pronounced at C band but also significant at X band (Trömel et al., 2013). Once the contribution of δ is removed, $K_{DP}$ is estimated by calculating the range derivative of $\phi_{DP}$ . Uncertainties in identifying the contribution of δ affects, however, the $K_{DP}$ estimates.”**

**The aerosol properties that need to specified in the model is better described in the revised manuscript:**

**Ln 105: ”The activation of CCN from aerosols in SB2M is based on pre-computed activation ratios stored in a lookup table (Siefert et al., 2012), which depend on vertical velocity and background aerosol properties (Segal and Khain, 2006). The aerosol is assumed to be partially soluble with a two-mode lognormal size distribution. This requires the specification of the condensation nuclei (CN) concentration, the mean radius of the larger aerosol mode, the logarithm of its geometric standard deviation, and its solubility.”**

**The aspect ratio and the canting angle distribution is now better described in the Appendix:**

**“The aspect ratio is the ratio between the horizontal and the vertical dimension of a particle. The canting angle is the angle between the horizontal and the symmetry axis of the falling particles (horizontally aligned particles have a 0° canting angle). In a radar observed volume containing several particles, canting angles vary from particle to particle giving rise to a distribution. The width of the canting angle distribution is a measure of the variability of canting angles in that sample.“**

3) In sections 1 and 2, authors cite too many online references or unpublished (not peer-reviewed) articles. The reviewer is skeptical with some of the research results mentioned in the paper. Please correct them or update them to more recent peer-reviewed papers.

**The overview article by Trömel et al. (2021) has been accepted for publication in the meantime. The reference Xie et al. (2021) only refers to the availability of the Bonn radar forward operator applied in this study for the community. However, it has been already successfully applied also in the peer-reviewed paper by Heinze et al. (2017) and the aforementioned accepted paper by Trömel et al. (2021). We now included Heinze et al. (2017) and Trömel et al. (2021) as an additional reference for the forward operator in Section 2.2. We also made use of the Bonn forward operator in Shrestha et al. (2021), which is currently in review for publication (GMD Discussion) in the same Special Issue (SI). To our knowledge, it is accepted to mention manuscripts currently in review for the same SI, but if the reviewer prefers, we will remove this reference.**

4) This study uses TSMP to study the evaluation of modeled summertime convective storms using polarimetric radar observations. However, in sections 3.1 and 3.3 the paper discusses about land cover types and root zone soil texture without any context. Based on the three cases which are only for short convective periods, the reviewer finds little purpose of the coupling nor its relation to dual-pol study.

In addition, lines 201-205, evaluation of streamflow and discharge in the model has little to do with the research purpose. Again, section 5.2 river discharge serve little purpose in the paper. The locations of streams are not even shown in Figure 3, which makes it impossible to understand the results.

The reviewer suggests either to change the title and research purpose or delete the above mentioned descriptions.

**We agree that the model evaluation of streamflow has little contribution to the main research work presented in the study. Following the reviewer's suggestion,**

**it has been now removed in the revised manuscript. Further, the discussion about the land cover type and soil texture is related to the model domain. It has been shortened to address the reviewer's additional concerns. Particularly, the discussion about root zone soil texture and aquifers has been removed.**

5)      Section 4.1 discuss synoptic situations of three cases but the reviewer finds it difficult to follow the descriptions. Some figures depicting synoptic conditions are needed.

**In the revised manuscript, we have added a new plot of surface pressure reduced to mean sea level and 850 hPa pseudo-equivalent potential temperature for the three cases based on Global Forecast System (GFS) model analysis at 12 UTC. Also, additional synoptic plots are also directly available from http://www1.wetter3.de . Following text has been added to the revised manuscript:**

**Ln 226: "Figure 2 shows the synoptic conditions for the three cases; shown are the surface pressure reduced to mean sea level and pseudo-equivalent potential temperature based on GFS analysis at 1200 UTC. Additional synoptic plots are also directly available from http://www1.wetter3.de. "**

6) The description in lines 260-262 is incorrect. Observations do not gradually increase as modeled CAF. In figure 4, there is no explanation of what (a), (b), (c) are. Further, the snapshot time periods (boxed area) between observations and model runs are different. The paper does not mention this issue or provide reasons of different time period selection. "Optimal" is not enough.

**The sentence has been rephrased in the revised manuscript:**

**Ln 259: "For the second case, CAF gradually increases for all ensemble members and remains quasi-steady after 1100 UTC. However, all ensemble members underestimate CAF in the earlier phase of the storm (before 1100 UTC) compared to observations."**

**Also, Figure 4 (Fig. 3 in the revised manuscript) has been improved to explain that a), b) and c) belong to Case One, Two and Three, respectively.**

**Comparing models with observations is always challenging due to mismatches of the simulated and observed storm evolution in space and time (also shown by the variability in the CAF evolution). So, besides exploring the time series of CAF, we also conducted a qualitative exploratory analysis (using synthetic polarimetric variables at lower level (~ 1000 m a.g.l.), mid-level (near melting layer), and upper level (2.5 km above melting layer) to find that simulated convective storm evolution interval closest in time and location to the polarimetric observations. Based on both analyses, we identified the suitable ensemble members, times(identified by square markers), and time intervals (solid lines bounded by vertical bars) for the comparison of the statistical distribution of polarimetric variables between observations and simulations.**

**The following paragraph has been added in the revised manuscript to address the reviewer's concerns:**

**Ln 283: "The comparison of model with observation is always challenging, due to mismatches of the simulated and observed storm evolution in space and time (also shown by the variability in the CAF evolution). So, besides exploring the time series of CAF, we also conducted a qualitative exploratory analysis (using synthetic polarimetric variables at lower levels (~ 1000 m a.g.l.), mid-levels (near melting layer), and upper levels (2.5 km above melting layer) to find the simulated convective storm among the ensemble members that was closest in time and location compared to the polarimetric observations. Based on the above two analyses, we identified the ensemble members, time-snapshot (identified by square markers) and time intervals (solid lines bounded by vertical bars) for the comparison of the statistical distribution of polarimetric variables between observations and simulations."**

7) In sections 5.3.1, 5.3.2, and 5.3.3, there are many leaps in steps and logic in terms of how locations, elevations, and time is determined for analyses (figures 5 – 10). How can you compare the radar signatures with model simulations when the locations, elevations, and time periods are different? The range plots in Fig. 5 and 6 are also different and very difficult to compare. In Fig. 9 why is 8.2 degree elevation used?

**Comparing a model with observation is always challenging, as models often have a shift in the actual location where convection develops, and/or a shift in the time of the development of the storm. What we did was to find the simulated convective storm evolution interval that was closest in time and location to the observations. We have added a paragraph in Section 5.3 to clarify the steps and logics. See also our comments above**.

**As for the range plots in Figs. 5 and 6, again, we are comparing polarimetric observations with simulated polarimetric data in their native coordinates. The dotted gray circles in Fig. 5 represent slant ranges for the chosen elevation angle, associated with heights of 1 km (lower levels) , 4.5 km (melting layer) and 7 km (upper layer); this is now clearly stated in the caption of the revised manuscript. Fig. 6 shows the simulated polarimetric variables at 1 km and 4.5 km. In the description of the plots, we focus on convective features present in both simulations and observations at these levels, and these are the important points shown in those images.**

**Volume scans consisting of a series of PPIs from different elevations, mostly between 0.5° and 30°, became more popular in recent years in order to get a 3D picture of hydrometeors and microphysical processes. Volume scans also allow for vertical cross-sections (e.g. Fig. 5b). A PPI at 8.2° elevations(Figs. 9a and 5a) better allows us to compare observations at different heights (~1 km, near melting layer and 2~3 km above melting layer) because the radar measures at increasing height with increasing distance from the radar. Together with the spatial extent and location of the system this complements the cross-section in Fig. 5b and Fig. 9b.**

**For clarity on the use of the different elevation scans, we have added the following text in Section 3.3 of the revised manuscript:**

**Ln 182: "Both X-band Doppler radars produce volume scans consisting of a series of ensuing Plan Position Indicators (PPIs) measured at different elevations, mostly between 0.5° and 30°. The use of these multiple sweeps became more popular in recent years in order to get a 3D picture of surrounding hydrometeors and microphysical processes. These PPIs can be exploited for improved process understanding, model evaluation and data assimilation. And, such volume scans also enable us to construct vertical cross-sections of convective systems."**

**Ln 211: "Based on the time and location of the storm from the radar, PPI measured at different elevation for each case are used, giving insights of the measurement of convective systems at different heights (~1 km, near melting layer and 2~3 km above melting layer)"**

8) Captions in figures 2, 4, 5-10 need to be improved. For instance, box plots of ensemble members in Fig. 2 are not explained well.

**The captions have now been improved in the revised manuscript.**

9) The first sentence in section 6 (discussion) is improperly placed. Further, in order to address the change in IC/BC and its influence on the simulated cases, the experiment needs to be conducted on the same cases. But, this paper does not. Thus, lines 424-429 should be deleted.

**In this study, we conduct ensemble simulation using 20 members for each case study. The ensemble was generated by using 20 different IC/BC based on COSMO-DE\* EPS data provided by DWD.**

**The COSMO-DE EPS data can be divided into 4 subsets of 5 members each. The 4 subsets represent different lateral boundary conditions obtained from global models. Each subset of the 5 members is then perturbed by varying a set of parameters that control the physics parameterization of the COSMO model.**

**\*COSMO-DE is a high resolution (~2.8 km) configuration of the COSMO model encompassing the entire extent of Germany.**

**The description of the COSMO-DE EPS data in Section 3.2 has been improved in the revised manuscript for clarity.**

**Minor comments (technical corrections):**

1) There are many acronyms used in this study without spelling them out properly (e.g.EMVORADO, DE, GME, JUWELS, a.g.l., etc.

**The full form of the missing acronyms has been added.**

2) There are many typos and unnecessary use of "e.g." used in this paper. Please correct them.

**Fixed.**

3) L59: There are many more recent X-band study results that are published. Please include them as references.

**We have now included additional references for recent studies that document polarimetric signatures of convective storms using X-band radar in the revised manuscript.**

**Allabakash, S., Lim, S., Chandrasekar, V., Min, K. H., Choi, J., & Jang, B. (2019). X-Band Dual-Polarization Radar Observations of Snow Growth Processes of a Severe Winter Storm: Case of 12 December 2013 in South Korea, Journal of Atmospheric and Oceanic Technology, 36(7), 1217-1235.**

**Das, S.K., Hazra, A., Deshpande, S.M. et al. Investigation of Cloud Microphysical Features During the Passage of a Tropical Mesoscale Convective System: Numerical Simulations and X-Band Radar Observations. Pure Appl. Geophys. 178, 185–204 (2021)**

**Trömel, S., Simmer, C., Blahak, U., Blanke, A., Ewald, F., Frech, M., Gergely, M., Hagen, M., Hörnig, S., Janjic, T., Kalesse, H., Kneifel, S., Knote, C., Mendrok, J., Moser, M., Möller, G., Mühlbauer, K., Myagkov, A., Pejcic, V., Seifert, P., Shrestha, P., Teisseire, A., von Terzi, L., Tetoni, E., Vogl, T., Voigt, C., Zeng, Y., Zinner, T., and Quaas, J.: Overview: Fusion of Radar Polarimetry and Numerical Atmospheric Modelling Towards an Improved Understanding of Cloud and Precipitation Processes, Atmos. Chem. Phys. Discuss. [preprint]**

4) L75: Runge-Kutta is not a "dynamical core" but "numerics" to solve PDE equations.

**Yes. The sentence has been rephrased for clarity:**

**Ln 83: "The dynamical core of COSMO uses the two time-level, third order Runge–Kutta method to solve the compressible Euler equations (Wicker and Skamarock, 2002; Baldauf et al., 2011)."**

5) L90: No à No_x

**To avoid any confusion, we have replaced $N_0$ with A in the revised manuscript.**

6) L175: The two X-band radars used in this study are calibrated based on GPM DPR (Ka band). However, there are many literatures that show ground based and airborne radar signatures are much different in characteristics and have lots of biases. Please provide some evidence of how DPR can be used as reference and not the other way around.

**We have a high confidence in the calibration based on GPM DPR (Ku-band), because they are consistent with results obtained with the methodology described in Diederich et al. (2015a), which is based on the consistency of polarimetric measurements of the ground-based radar only. As an additional cross-check, we also create Quasi-Vertical-Profiles after application of the offsets determined to see whether in rain the differential reflectivities are in line with the values to be expected given the reflectivity measurements (reflectivity $Z_H$ and differential reflectivity $Z_{DR}$ show a positive correlation in rain). Relationships for X, C, and S-band are shown in the book by Ryzhkov and Zrnic (2019).**

**Furthermore, the calibration technique selects only stratiform events where a bright band is visible, and only reflectivities between 10 dBZ and 36 dBZ are taken into account, to avoid strong effects of attenuation.**

**Successful calibrations of ground-based radars with satellite-based radars have been done in several previous studies:**

**Warren, R. A., Protat, A., Siems, S. T., Ramsay, H. A., Louf, V., Manton, M. J., & Kane, T. A. (2018). Calibrating Ground-Based Radars against TRMM and GPM, *Journal of Atmospheric and Oceanic Technology*, *35*(2), 323-346.**

**Protat, A., D. Bouniol, E. O. Connor, H. Klein Baltink, J. Verlinde, and K. Widener, 2011: *CloudSat* as a global radar calibrator. *J. Atmos. Oceanic Technol.*, 28, 445–452.**

**Louf, Valentin, et al. "An integrated approach to weather radar calibration and monitoring using ground clutter and satellite comparisons." Journal of Atmospheric and Oceanic Technology 36.1 (2019): 17-39.**

**Crisologo, I., Warren, R. A., Mühlbauer, K. & Heistermann, M.  Enhancing the consistency of spaceborne and ground-based radar comparisons by using beam blockage fraction as a quality filter.Atmospheric Meas. Tech.11,  5223–5236,218 (2018).**

**Schwaller, M. R., and K. R. Morris, 2011: A ground validation network for the Global Precipitation Measurement mission. J. Atmos. Oceanic Technol., 28, 301–319.**

**The following discussion has been added in the revised manuscript:**

**Ln 189: "The calibration based on GPM DPR (Ku-band) is consistent with results obtained with the methodology described in Diederich et al. (2015a). Furthermore, the calibration technique selects only stratiform events where a bright band is visible, and only reflectivities between 10 dBZ and 36 dBZ are taken into account, to avoid strong effects of attenuation. Successful calibrations of ground-based radars with satellite-based radars have been also been done in several previous studies (Protat et al. 2011; Schwaller and Morris 2011; Warren et al. 2018; Crisologo et al. 2018; Louf et al. 2019;)"**

7)      L224: How does anticyclonic rotation of the warm front produce the necessary lifting mechanism?

**The sentence has been rephrased for clarity:**

**Ln 244: "This additional northward push of the warm front ...."**

8)      L231: Between "ensemble average." and "For the first case," there needs to be a sentence explaining Figure 2.

**We have added the following sentence in the revised manuscript:**

**Ln 249: "Figure 3 shows the spatial pattern and frequency distribution of the modeled and observed accumulated precipitation over the Bonn Radar domain for the three case studies."**

9)       L237: northwestern domain à northeastern domain

**Corrected.**

10)      L278: How can you tell where the melting layer is based on Fig. 5a. Explain in more detail.

**The dotted gray circles represent slant ranges associated with heights of 1 km (lower levels) , 4.5 km (melting layer) and 7 km (upper levels). The melting layer height is identified from the temperature profile of model output. This has been now better explained in the Figure caption of the revised manuscript.**

---

## Author Response (AR2)

**Editor comments to the author:**

I have now (finally) two re-reviews of your paper. I apologize again for the delay in obtaining these reports.
As you see, the comments on your paper are rather different. I also note that some of the issues (e.g. literature) could have been raised earlier. Nonetheless, I would like to ask you to address the comments of rev. #3 in a revised version. Perhaps you could make the "way forward" and "what is new" clearer in your paper.

**We are very thankful for the editor's comments. Below, we address the reviewer #3's specific comments (in bold blue) and make an effort to present the "way forward" and "what is new" more clearly in the revised manuscript.**

**Reviewer #3**

Overarching comments

This study examines three case studies of convective storms observed with two X-Band radars in Germany, comparing synthetic polarimetric observations simulated by an ensemble of convective permitting model simulations with observations. The forward operator is applied to the model output with a scheme to correct for propagation effects at X-Band (attenuation, differential attenuation). The study compares the evolution of surface rain rates, convective area fraction, selected time snapshots of polarimetric variables and hydrometeor mixing ratios, and time and space averaged cumulative frequency by altitude diagrams (CFADs) of polarimetric variables to conclude potential shortcomings of the coupled model simulations and forward operator technique.

While the end-to-end forward simulation of polarimetric observables from radar is indeed a potential pathway to diagnose model shortcomings (which is major goal of this study), the findings are qualitative in the sense that due to the necessary compromises in sampling the synthetic observations and the observations themselves that may suffer from significant sampling and representativeness biases, as well as errors due to the application of the forward operators. The authors note some of these potential uncertainties, but do not deal with them. Accounting for these uncertainties, it is not clear what the path forward is in understanding the issues with the model (is there a problem with the kinematics/thermodynamics/microphysics, or is it a shortcoming of the model setup or initial conditions?) towards ultimately improving the state of the art of cloud modeling. Can the authors specify the path forward?

**We very much agree with the reviewer that there is a considerable knowledge gap in the fusion of radar polarimetry with atmospheric models. The uncertainties in the assumptions made in the forward operator (FO) and attenuation correction of observations really challenge and question the effectiveness of the model evaluation approaches with polarimetric radar data. However, we have to note here that the assumptions made in the FO**

and attenuation corrections applied to the observations are based on the state of art knowledge. We acknowledge these limitations to exercise caution while interpreting the model evaluation, and thereby focus more on patterns to directly evaluate the shortcomings on the modeled signatures of the microphysical processes.

While ensemble simulation for multiple convective storms carried out in this study already shows the importance of initial and lateral boundary conditions for the improvement of simulated precipitating cloud system, a way forward would be to have additional ensemble FO with a combination of hydrometeor parameters to explore the full spread of synthetic polarimetric moments. Earlier, an extensive sensitivity study with the hydrometeor parameters in the FO was also conducted for a stratiform case over the same modelling domain (Shrestha et al. 2021). The model was found to exhibit a low bias in the polarimetric moments above the melting layer, where snow was found to dominate, but none of the alternative shape and orientation setups for snow could provide sufficiently strong polarimetric signals to reproduce observed signals at these heights. This sensitivity study, thus helps to point towards missing shortcomings in the cloud microphysical scheme and/or the scattering estimates using T-matrix calculations. Such low biases above the melting layer are also observed in this study.

Besides, evaluation of synthetic patterns of polarimetric signatures with observation (e.g., $Z_{DR}$ /$K_{DP}$ columns) already helps in identifying the possible deficiencies in the cloud microphysical schemes and the FO. So, in addition to ensemble FO, model evaluation should particularly focus on the dominant polarimetric signatures of the precipitating cloud systems. The model was particularly found to underestimate convective area fraction, width and magnitude of $Z_{DR}$ columns, associated with small sized supercooled raindrops, which could be a result of the fixed CN concentrations and shape parameters of cloud drop size distribution, and the water content determination of the ice hydrometeors in the FO.
So, either the use of regional measurements of CN/IN concentrations, or sensitivity study with large scale aerosol perturbations or use of prognostic aerosol/trace gasses module could be a way forward to minimize the uncertainty in polarimetric signatures due to aerosols. Therefore, sensitivity studies with different parameters for cloud activation/ice nucleation and direct use of a chemistry transport model for prognostic aerosol simulation for the same case study have also been undertaken and are in the process of being submitted. Besides, work also needs to be undertaken to develop additional parameterization in the FO to determine the water content of the ice hydrometeors above the melting layer (e.g., water content for wet growth of hail etc). Furthermore, the modelling and observational community need to work more closely together (as fostered in PROM; Trömel et al. 2021) to achieve the above objectives.

The following text has been added in the revised manuscript:
Ln 567: "For the 2-moment cloud microphysics scheme, the fixed CN concentrations and shape parameters of cloud drop size distribution could also be partly responsible for the overall too low storm intensities, thus regional measurements of CN/IN concentrations, surface precipitation and polarimetric radar data observations could be used together to

constrain the shape parameters of cloud droplets. While regional measurements of CN/IN concentrations might not be readily available, sensitivity study with large scale aerosol perturbations or use of prognostic aerosol/trace gasses module could be a way forward to minimize the uncertainty in polarimetric signatures due to aerosols. On the forward operator for 2-moment cloud microphysics scheme, the water content of the ice hydrometeors can strongly modulate the dielectric constant and hence the scattering properties. This information is not directly available in the forward operator - and the melting parameterization in the FO does not completely compensate for the scattering properties of the ice hydrometeors above the melting layer. So, future advancement in the FO should include parameterization for determining more accurate water content of the ice hydrometeors above the melting layer, which would help in obtaining more accurate dominant polarimetric signatures. Importantly, the prominent polarimetric signature of convective storms like the $Z_{DR}$ column appears to be poorly resolved at km-scale simulations. Future model evaluations with polarimetric radar data should focus on hyper-resolution simulations to better resolve the three-dimensional motion and microphysical processes associated with multivariate polarimetric signatures as well as uncertainty estimates in the attenuation correction of polarimetric moments for convective cases."

Furthermore, the authors note large discrepancies between the snapshots comparisons as well as the statistical comparisons using CFADs. The reviewer notes that this approach has been taken in prior work (Pfeifer et al. 2008; Kumjian et al. 2019; Matsui et al. 2019), which leads to the logical question about what is new here? The present work does not present itself as a novel advance of the state of the topic under investigation. Nor does it present an improvement to methodology, the state of the art in modeling, nor the use of radar data. Thus, the reviewer has serious concerns about the need to publish this paper in the literature. I suggest that the authors take some time in revision to state what is new in this paper.

We are thankful for the reviewer's suggestion of previous works which have taken a similar approach for model evaluation with radar polarimetry. We have included these references in the revised manuscript also, and compared them in context to the findings from this study.

Even though first polarimetric forward operators have already been available several years ago, like SynPolRad introduced in Pfeifer et al. (2008), there is still a considerable knowledge gap in the fusion of radar polarimetry with atmospheric models. The acceptance of PROM project as a special priority programme by the German Research Foundation confirms the novelty and great potential exploiting radar polarimetric observations for a more detailed model evaluation and ensuing improvements (see PROM-overview paper: Trömel et al. 2022)- refinements of forward operator are still ongoing and mandatory for a full exploitation (like connection to scattering data base for better representation of the ice phase) and their full exploitation for model evaluation is still at the beginning, partly because polarimetric precipitation radar networks became just recently available (in Germany since 2016) as well as newly developed tools (e.g. QVPs). Furthermore, each atmospheric model has their own cloud microphysical schemes with different levels of sophistication. In this study, we use the Terrestrial Systems Modeling Platform to perform km-scale ensemble simulations in convection permitting mode, to evaluate 2-moment cloud

microphysics scheme (Seifert and Beheng 2006; hereafter SB2M) for multiple convective storms using a FO with high resolution X-band polarimetric radar data. The 2-moment scheme allows the possibility of aerosol-cloud-precipitation interaction studies and hence the possibility of aerosol effects on polarimetric quantities. The findings from this study points to such possible connections, which will be addressed in an upcoming sensitivity study with cloud activation and ice nucleation parameters. Further, the SB2M scheme is also a candidate for the ICON model used for operational weather forecasting by DWD. And, previous studies have mostly documented polarimetric signatures of convective storms in C or S band, while studies based on high resolution X-band are still gaining ground.

To particularly address the reviewer's concerns, the Introduction section has been revised to better clarify the motivation for this research. Importantly, we agree with the reviewer that the findings should be made clearer and associated/interpreted together with our challenges to present what is new in this study. This has been now discussed and presented more clearly in the revised manuscript in context to polarimetric signatures and statistical distributions of polarimetric variables.

[revised manuscript text omitted]

Specific comments

L133: remove "for those"

**Done.**

L228: "connected to" should be "associated with a"

**Corrected.**

L233: lightnings should be singular

**Corrected.**

L236: rephrase "connected with" again.

**Done.**

L241: this sentence should be rephrased or removed. It is sufficient to say that a warm front extended across central Germany. The "wave like feature" is also not apparent in Figure 2c.

**The sentence has been removed.**

L252: Do you quantify "gradient" or just accumulation frequency?

**We only quantify the frequency distribution of accumulated precipitation. The sentence has been rephrased for clarity.**

**Ln 275: "Overall, the spatial pattern of ensemble averaged accumulated precipitation resembles the RADOLAN estimates, but the frequency distribution produced by the ensemble members underestimate high precipitation."**

L272: It is unusual to start a sentence with "And"
**Removed and rephrased.**

Figure 5: Label with Case 1 as in Fig 4.
**Added in caption.**

L332: "hails" should be "hail"
**Fixed.**

L338-9: L352: "upto" should be two words.
**Fixed.**

L475: "shift" -> "shifts"
**Fixed.**